**EMBO** *reports*

# Type I interferon exacerbates *Mycobacterium tuberculosis* induced human macrophage death

Angela M Lee [ID][1,2] & Carl F Nathan [ID][1,2 ✉]

## Abstract

Type I interferons (IFN-I) are implicated in exacerbation of tuberculosis (TB), but the mechanisms are unclear. Mouse macrophages infected with *Mycobacterium tuberculosis* (Mtb) produce IFN-I, which contributes to their death. Here we investigate whether the same is true for human monocyte-derived macrophages (MDM). MDM prepared by a conventional method markedly upregulate interferon-stimulated genes (ISGs) upon Mtb infection, while MDM prepared to better restrict Mtb do so much less. A mixture of antibodies inhibiting IFN-I signaling prevents ISG induction. Surprisingly, secreted IFN-I are undetectable until nearly two days after ISG induction. These same antibodies do not diminish Mtb-infected MDM death. MDM induce ISGs in response to picogram/mL levels of exogenous IFN-I while depleting similar quantities from the medium. Exogenous IFN-I increase the proportion of dead MDM. We speculate that Mtb-infected MDM produce and respond to minute levels of IFN-I, and that only some of the resultant signaling is susceptible to neutralizing antibodies. Many types of cells may secrete IFN-I in patients with TB, where IFN-I is likely to promote the death of infected macrophages.

**Keywords** Interferon; Macrophage; Mycobacterium; Tuberculosis; Cell Death
**Subject Categories** Immunology; Microbiology, Virology & Host Pathogen Interaction; Signal Transduction

## Introduction

*Mycobacterium tuberculosis* (Mtb) is estimated to have infected about a quarter of the world's population (Houben and Dodd, 2016). An estimated 5–10% of infected individuals develop active tuberculosis (TB), which caused ~1.6 million deaths in 2021 (World Health Organization, 2022). Type I IFNs (IFN-I) may play a key role in promoting the progression to active disease. Patients with TB display a strong IFN-I response gene signature compared to healthy individuals with and without immunologic evidence of Mtb infection (Berry et al, 2010; Roe et al, 2016; Singhania et al, 2018; Tabone et al, 2021). ISGs were among the earliest genes to be upregulated in blood cells from individuals who were progressing to active TB (Scriba et al, 2017). Administration of IFN-I to treat hepatitis C or multiple sclerosis has led to the development of active TB in patients with latent TB infection (Belkahla et al, 2010; Matsuoka et al, 2016; Sabbatani et al, 2006; Sirbu et al, 2020), and TB patients with genetic deficiencies in IFN-I signaling were reported to have better clinical outcomes than those with an intact IFN-I signaling pathway (Zhang et al, 2018). Mtb burden and pulmonary pathology were reduced and survival was extended in mice in which IFN-I signaling was genetically disrupted or immunologically blocked (Dorhoi et al, 2014; Thirunavukkarasu et al, 2023; Zhang et al, 2020). In mice infected with Mtb, adverse effects of IFN-I on the host were traced to its actions on interstitial pulmonary macrophages, but what the adverse effects consisted in was not resolved (Kotov et al, 2023).

One way in which IFN-I may promote Mtb pathogenesis is by mediating macrophage cell death (Zhang et al, 2020). Most of the intracellular Mtb in a host with TB is contained within monocyte-derived macrophages (MDM) that have been recruited to the sites of infection (Huang et al, 2019; Lee et al, 2020), and Mtb replicates exponentially in necrotic macrophages (Lerner et al, 2017; Mahamed et al, 2017). Mathematical modelling predicted that the rate of macrophage apoptosis is the most significant factor for progression from latent TB infection to active TB (Zhang et al, 2021). In the lungs of non-human primates infected with Mtb, 15–60% of myeloid cells were "IFN-responsive macrophages" displaying a strong IFN-I and -II transcriptional signature (Esaulova et al, 2021). Higher presence of IFN-responsive macrophages positively correlated with increased levels of Mtb colony forming units (CFUs). A genome-wide CRISPR-Cas9 screen in a mouse macrophage cell line implicated components of the IFN-I signaling pathway in Mtb induced macrophage cell death (Zhang et al, 2020). Knocking out the IFN-I receptor (IFNAR) or blocking IFNAR with antibodies reduced or delayed the Mtb-induced death of mouse bone marrow derived macrophages (BMDM) (Zhang et al, 2020) and alveolar macrophages (Dorhoi et al, 2014), especially if the BMDM were primed with IFNγ (Zhang et al, 2020).

Both mouse BMDM and human monocytic cell lines secrete IFN-I in response to Mtb infection (Danelishvili et al, 2012; Zhang et al, 2020) via cGAS-STING activation (Manzanillo et al, 2012; Wassermann et al, 2015). Binding of IFN-α or IFN-β to IFNAR initiates JAK1/TYK2-STAT1/2 phosphorylation and activation, resulting in the upregulation of IFN-stimulated genes (ISGs)

[1]Department of Microbiology & Immunology, Weill Cornell Medicine, New York, NY 10065, USA. [2]Immunology & Microbial Pathogenesis Program, Weill Cornell Graduate School of Medical Sciences, New York, NY 10065, USA. ✉E-mail: cnathan@med.cornell.edu

(Ivashkiv and Donlin, 2014). Surprisingly, however, it is unclear whether primary human MDM secrete IFN-I upon Mtb infection. Novikov et al (2011) observed transcriptional upregulation of *IFNA1* and *IFNB* in human MDM infected with Mtb but did not report whether IFN-I proteins were secreted. Giacomini et al (2001) found no secretion of IFN-α by Mtb-infected human MDM but did not measure whether IFN-β was secreted, nor whether *IFNA/B* were upregulated. Mayer-Barber et al (2014) reported IFN-β in the supernatants of Mtb-infected human MDM but did not compare whether more IFN-β was released by Mtb-infected human MDM than by uninfected MDM, and the reported concentration of secreted IFN-β may have been below the limit of detection described by the manufacturer of the assay used. Thus, it is unclear if IFN-I is just transcriptionally upregulated by Mtb-infected human MDM or if they also secrete IFN-I.

Complicating comparison of results, multiple protocols are used for the differentiation of macrophages in vitro from human blood monocytes, some of which are compared in Table 1. For example, Novikov et al (2011) and Mayer-Barber et al (2014) stimulated monocytes with macrophage colony stimulating factor (M-CSF), while Giacomini et al (2001) used granulocyte-macrophage colony stimulating factor (GM-CSF). Both Novikov et al (2011) and Giacomini et al (2001) cultured the human MDM in RPMI 1640 supplemented with FBS for 5–7 days, presumably at atmospheric levels of oxygen, while the type of medium and possible use of cytokines by Mayer-Barber et al (2014) were not reported. In 2011, Vogt and Nathan (2011) described an extensive investigation of culture parameters that optimized human MDM cell viability upon Mtb infection at low multiplicities of infection (MOI) while minimizing Mtb growth within the MDM. This more cumbersome method involved use of more physiological conditions, such as human plasma and incubation at tissue levels of oxygen, along with combined exposure to GM-CSF and TNFα followed by IFNγ, all at concentrations below those used in most studies.

Here, to resolve the ambiguity whether human MDM secrete IFN-I in vitro in response to Mtb infection and to determine whether IFN-I signaling contributes to the Mtb-induced death of human MDM, we employed two methods to differentiate MDM: MDM-1, as described by Novikov et al (2011), which resembles methods used in most reports on human MDM; and MDM-2, developed by Vogt and Nathan (2011), which was developed specifically to minimize Mtb-induced death of human MDM without consideration of the possible role of IFN-I. Hereafter, we refer to the MDM by the name of the method used to prepare them.

Neither method led to detectable IFN-I secretion at a time when ISGs were induced in MDM-1, and MDM-2 barely expressed ISGs. The effects of antibodies that neutralize effects of added IFN-I differed depending on which responses were measured, ISG upregulation or IFN-I-mediated Mtb-induced cell death. Finally, exogenous IFN-I exacerbated Mtb-induced cell death of both MDM-1 and MDM-2. Therefore, in the human host, where TB can lead to IFN-I secretion by many types of cells, IFN-I is likely to contribute to death of MDM.

## Results

### Anti-IFN-I neutralizing antibodies do not inhibit Mtb-induced MDM cell death

Because anti-IFNAR antibodies blocked Mtb-induced cell death of mouse bone marrow derived macrophages (Zhang et al, 2020), we began these studies with the assumption that the same was likely true for human MDM. Infection with Mtb H37Rv at multiplicity of infection (MOI) of 5 or 20 led to death of both MDM-1 and MDM-2, as determined by residual ATP levels normalized to the uninfected, media values (Fig. 1A,B) and microscopy (Fig. EV1A). We identified a mixture of anti-IFN-I and anti-IFNAR antibodies (Abs) that completely inhibited STAT1 phosphorylation upon IFN-β stimulation in MDM-1 and MDM-2 (Fig. EV1B–E). The neutralizing IFN-I Ab mixture specifically inhibited signaling by IFN-I and not by IFN-γ or type III IFN (IFN-III) (Fig. EV1E,F). However, there was no statistically significant difference in the level of Mtb-induced cell death between MDM treated with the neutralizing IFN-I Abs and MDM treated with an IgG2a isotype control or vehicle control (Fig. 1A,B; Appendix Figs. S1A and S1B). IFN-γ priming of both types of MDM did not change this phenotype (Fig. 1A,B). These results indicate that cell death of Mtb-infected MDM can occur in a manner that does not depend on the action of IFN-I in a form that is accessible to the extracellular neutralizing IFN-I Ab mixture used.

### Mtb-infected MDM do not secrete IFN-I detectable at the pg/mL level by 1 day post infection

The foregoing results led us to question if human MDM even secrete IFN-I when infected with Mtb. We infected MDM-1 and

**Table 1.    Differentiation methods used in studies of type I IFN production by human MDM.**

| Method | MDM-1 | MDM-2 | MDM-3 | MDM-4 |
|---|---|---|---|---|
| Citation | Novikov et al (2011) | Vogt and Nathan (2011) | Mayer-Barber et al (2014) | Giacomini et al (2001) |
| Cytokine | 10 ng/mL M-CSF | 0.5 ng/mL GM-CSF + TNF-a | 60 ng/mL M-CSF | 0.1 ng/mL GM-CSF |
| Media | RPMI-1640 + 10% FBS + 1% HEPES, + 1% L-glutamine (2 mM) + 50 μM β-ME | RPMI-1640 + 40% human plasma + Glutamax (2 mM) | Not stated | RPMI-1640 + 15% FBS + 1% L-glutamine (2 mM) |
| $O_2$ | 20% (presumably; not stated) | 10% | 20% (presumably; not stated) | 20% (presumably; not stated) |
| Differentiation period | 1 week | 2 weeks | 1 week | 5 days |
| CD14 monocyte isolation | Unclear | Yes | Unclear | Yes |

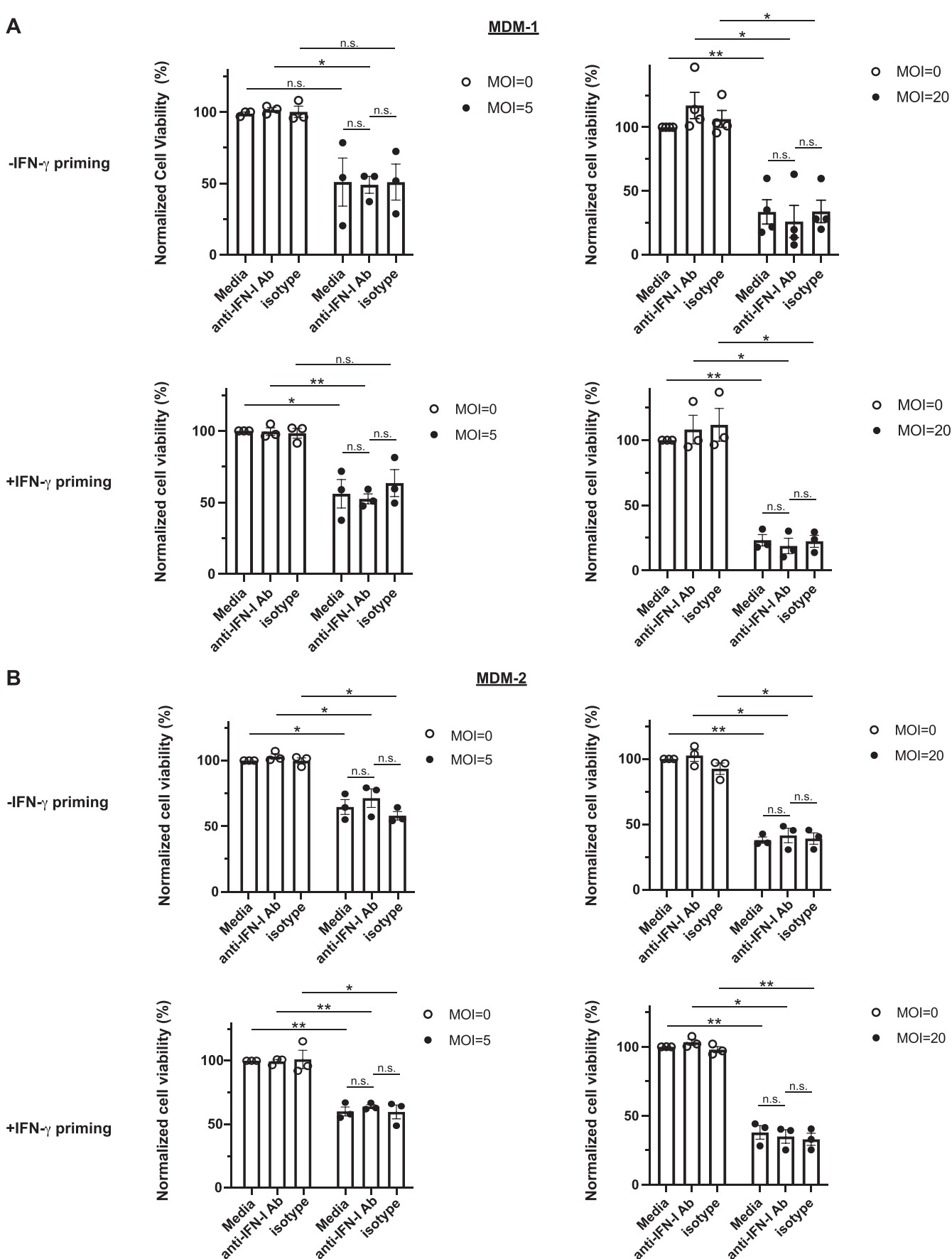

◄ **Figure 1. Anti-IFN-I neutralizing antibodies do not inhibit Mtb-induced death of MDMs.**

MDM-1 (**A**) or MDM-2 (**B**) primed with (bottom) or without (top) 2.5 ng/mL IFN-γ, treated with 1:1000 dilution of neutralizing antibodies (Ab) against IFN-I and IFNAR, isotype control Ab (IgG2a) or medium alone (vehicle control) for 2 h pre-infection, and infected with Mtb strain H37Rv. 4 h post infection, extracellular Mtb was washed out and medium was replaced, along with the respective Abs. Cell viability was measured 4 d post infection (left) or 2 d post infection (right). The cell viability was normalized to each donors' value for uninfected cells not exposed to antibodies. Data Information: In (**A and B**), data are presented as mean ± standard error of mean (SEM) of the normalized cell viability. Each ● or ○ represents the average value of 3 technical replicates per donor, $n = 3$ (except (**A**) top right: $n = 4$), while $n$ indicates biological replicates. Statistical significance was determined using a paired two-tailed, $t$-test (*$p < 0.5$; **$p < 0.01$; ***$p < 0.001$; n.s. indicates no statistical significance). Source data are available online for this figure.

MDM-2 with Mtb at an MOI of 5 or 20 and measured IFN-I at 1 day post infection using an IFN-I HEKBlue reporter assay that can detect IFN-β and all IFN-α subtypes. An increase in IFN-I at the pg/mL level was not detected in the supernatants of Mtb-infected MDMs compared to uninfected MDM (Fig. 2A). Results were the same using a high sensitivity IFN-β ELISA that has a limit of detection of 1.2 pg/mL (Fig. EV2A), and results were no different after priming with IFNγ (Fig. 2A). In contrast, and as reported (Collins et al, 2015; Danelishvili et al, 2012; Wassermann et al, 2015), the human monocytic leukemia cell line THP-1 and the histiocytic lymphoma cell line U937 differentiated with phorbol 12-myristate 13-acetate (PMA) did secrete detectable IFN-I upon Mtb infection (Fig. EV2A). Both MDM-1 and MDM-2 responded to Mtb by secreting copious TNFα by 1 day post infection (Figs. 2B and EV2B). MDM-1 did secrete IFN-I when treated with di-amidobenzimidazole (di-ABZI), which activates stimulator of IFN genes (STING) (Fig. 2C). In contrast, di-ABZI-stimulated release of IFN-1 by MDM-2 was not statistically significant, even though both MDM-1 and MDM-2 secreted TNFα 1 day post infection when treated with di-ABZI (Fig. 2C,D). Myeloid cells in tuberculous granulomas in cynomolgus macaques expressed IFN-III (Talukdar et al, 2022). Therefore, we also checked IFN-III secretion by MDM. However, neither MDM-1 nor MDM-2 secreted statistically significantly more IFN-III 1 day post infection or upon di-ABZI treatment than did uninfected or unstimulated MDM, respectively (Fig. EV2C,D).

Given that mice infected with different clinical Mtb strains isolated from human donors displayed differential cytokine expression in their lungs (Verma et al, 2019), we infected MDMs with two clinical strains of Mtb. Neither clinical strain induced MDM to secrete detectable IFN-I but both induced secretion of TNFα 1 day post infection (Fig. EV2E,F), consistent with the results for the laboratory strain, H37Rv.

To test whether Mtb inhibits IFN-I secretion by MDM, MDM-1, and MDM-2 were infected with Mtb H37Rv and stimulated with di-ABZI 4 h post infection. IFN-I proteins in the supernatant of MDM-1 and MDM-2 treated with di-ABZI were not reduced by infection with Mtb at a MOI of 5 or 20 (Fig. 2E,F), although there was a trend toward lower levels of IFN-I at the higher MOI.

### MDM-I upregulate ISGs in an IFN-I-dependent manner

Transcriptome-wide RNA sequencing on Mtb infected MDMs did not reveal upregulation of transcripts for IFN-I, -II, or -III at 24 h post infection (Fig. 3A). However, di-ABZI treated MDMs did not upregulate these transcripts either, despite copious secretion of IFN-I at this time point. Perhaps these transcripts were upregulated earlier and had returned to baseline by 24 h post infection, similar to what has been observed in other settings (Thomsen et al, 2022).

Nevertheless, MDM-1 and MDM-2 both highly upregulated ISGs upon di-ABZI stimulation (Figs. 3A and EV3A). MDM-1, but not MDM-2, upregulated ISGs upon Mtb infection to a similar level as di-ABZI treated MDMs (Fig. 3A). For Mtb-infected MDM-1, IFN-I signaling was among the 5 most highly upregulated pathways in the transcriptome (Fig. 3B). As expected, IFN-I signaling was the top pathway upregulated in the di-ABZI treated MDM-1 and MDM-2 (Fig. EV3B).

Because IFN-I proteins were not detected in the supernatants of Mtb-infected MDM-1 but ISGs were upregulated, we sought to determine if Mtb-infected MDM-1 upregulated ISGs in an IFN-I-dependent manner. Indeed, neutralizing IFN-I Abs ablated induction of the ISGs *IFITM3* and *ISG15* in Mtb-infected MDM-1 (Figs. 3C and EV3B).

### Differentiation time and amount or species of serum or plasma in the medium influence the level of IFN-I secretion upon STING activation and the level of ISG upregulation upon Mtb infection of MDM

We next compared features of the methods used to prepare MDM-1 and MDM-2 to try to understand their striking differences in expression of IFN-I upon STING activation and of ISGs upon Mtb infection. We differentiated MDM in RPMI 1640 with 40% human plasma and Glutamax at euoxic tissue levels of oxygen (10%) and exposed them either to exogenous M-CSF or to GM-CSF and TNFα for one or two weeks. When keeping the time of differentiation constant, M-CSF-treated MDM did not secrete IFN-I at greater levels than GM-CSF- and TNFα- treated MDM (Fig. 4A). In fact, after 1 week in culture, GM-CSF and TNFα treated MDM secreted more IFN-I upon STING activation than did M-CSF differentiated MDM (Fig. 4A). However, culture for 2 weeks diminished the levels of IFN-I secreted in response to STING activation of MDM exposed to either cytokine regimen (Fig. 4A). Thus, the difference in the level of IFN-I secretion by the MDM-1 versus MDM-2 is due in part to the length of differentiation time. For further experiments, we modified the MDM-2 protocol to limit the pre-infection culture time to 1 week. However, this did not result in comparable induction of *ISG15* in Mtb-infected MDM-2 as in MDM-1, even when the cytokine regime was switched to M-CSF (Appendix Fig. S2A). By inference, the use of human plasma rather than FBS was an additional reason for lower expression of ISGs in MDM-2. Indeed, when we kept all factors of the differentiation protocol the same for MDM-1 except for the type and amount of serum or plasma, MDM cultured in 40% autologous plasma failed to upregulate ISGs upon Mtb infection (Fig. 4B), although they responded to the artificial STING activator comparably after culture in 10% FBS or 40% autologous plasma (Fig. 4B).

In sum, compared to MDM-1, MDM-2 have a blunted ISG response to Mtb infection because they have been cultured in 40%

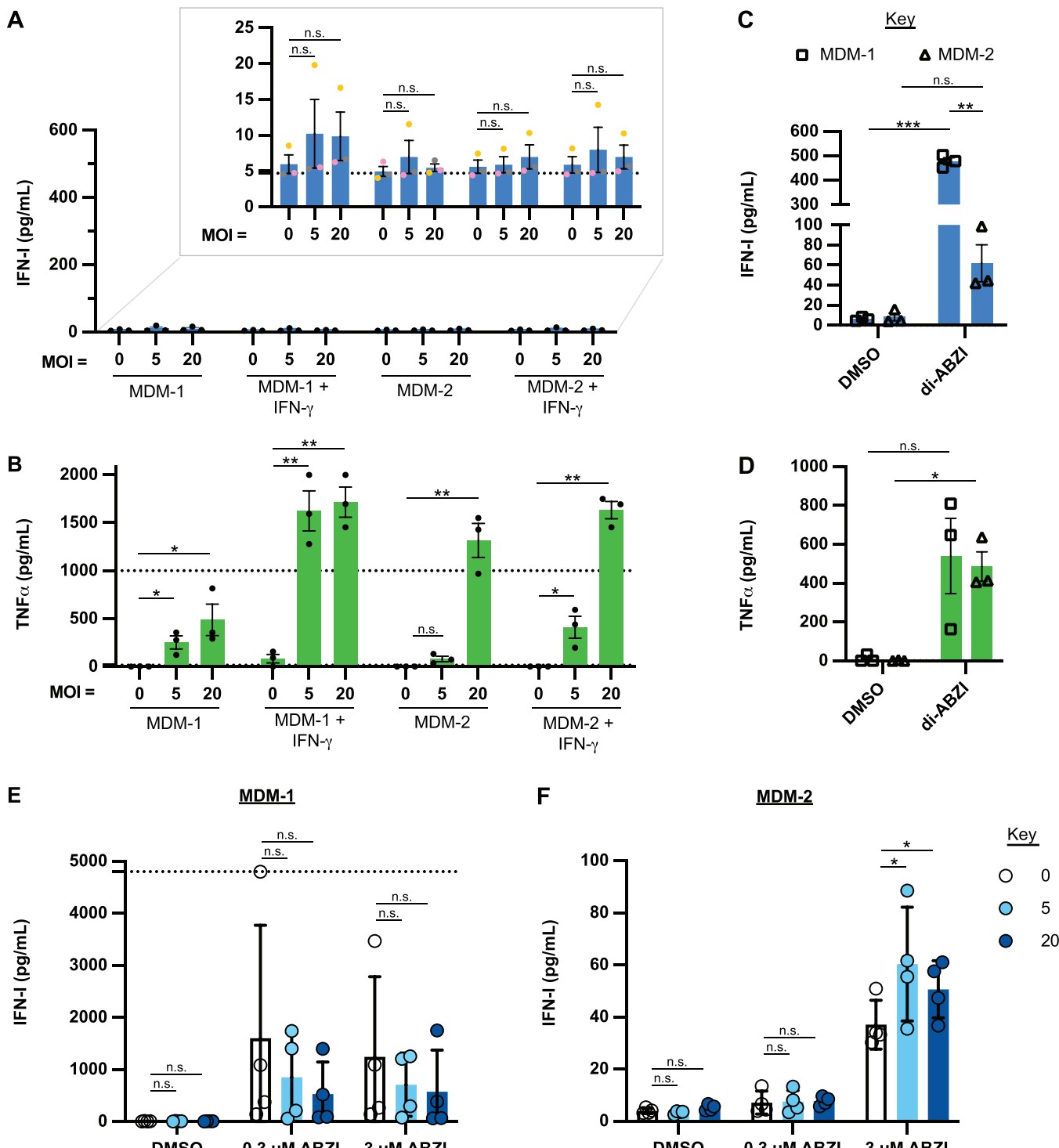

autologous plasma rather than 10% FBS and have been cultured longer than MDM-1. Those conditions led to striking morphologic differences as well (Fig. 4C). MDM-2 were much larger than MDM-1 (Fig. 4C) and had much greater levels of RNA (Appendix Fig. S2B), consistent with the marked increase in size reported for MDM differentiating in medium containing 25% human serum (Nakagawara et al, 1981).

## Mtb-infected MDM-1 upregulate ISGs before they secrete detectable IFN-I

Given that we did not detect IFN-I in the supernatant of Mtb-infected MDMs at 24 h post infection (Fig. 2A), we asked if IFN-I may have been secreted earlier but retained on the cells or otherwise removed from the extracellular medium. Accordingly, we

**Figure 2. Mtb-infected MDMs do not secrete IFN-I detectable at the pg/mL level at 1 d post infection.**

MDM-1 and MDM-2 (n = 3) were primed with or without 2.5 ng/mL IFN-γ for 1 day and then (A, B) infected with Mtb strain H37Rv or (C, D) treated with 3 µM di-ABZI or DMSO as a vehicle control (final DMSO, 0.06%). Supernatants were collected 1 d post infection or treatment. (A, C) IFN-I expression in the supernatant was measured with the IFN-I HEKBlue reporter assay and quantified by performing a sigmoidal, 4 parameter interpolation from an IFN-β standard curve. (B, D) TNFα expression was measured by ELISA. MDM-1 (E) and MDM-2 (F) (n = 4) were infected with Mtb H37Rv and treated with di-ABZI (0.3 µM or 3 µM) or vehicle control (DMSO, final concentration 0.06%) after Mtb was washed out at 4 h post-infection. Supernatants were collected 1 d post infection. IFN-I expression in the supernatant was measured by the IFN-I HEKBlue reporter assay and quantified by interpolating from an IFN-β standard curve. Data Information: (A–F) Bar graphs report the mean ± SEM. Each point represents the average value of 2–3 technical replicates per donor, while n indicates biological replicates. (B, D) Values too low to be interpolated were assigned a value of 1 pg/mL, and values too high to be interpolated were assigned a value of 2000 pg/mL. (E) Values too high to be interpolated were assigned a value of 4800 pg/mL. (A–F) Statistical significance was determined using a one-tailed paired, t-test (*p < 0.5; **p < 0.01; ***p < 0.001; n.s. indicates no significance), except for the difference between MDM-1 and MDM-2 stimulated with di-ABZI (C), which was determined by a two-tailed, paired t-test. Source data are available online for this figure.

measured IFN-I in the supernatant of Mtb-infected MDM-1 at 4, 8, 12, 18, 24, and 48 h post infection. Although not statistically significantly greater, there was a trend that IFN-I was first detectably greater in the medium at 48 h post infection compared to the uninfected cells (Fig. 5A), despite upregulation of *IFITM3* and *ISG15* by 8–12 h post infection (Figs. 5B and EV4A). Indeed, uninfected MDM-1 caused the disappearance of ~20% of 100 pg/ mL of exogenously added IFN-β within 4 h and about half of it within 8 h (Fig. EV4B). The amount bound and perhaps internalized by 4 h (20 pg/mL) was comparable to the amount (10 pg/mL) needed to induce *IFITM3* or *ISG15* when added to uninfected MDM (Figs. 5C and EV4C), but less than the amount of exogenous IFN-I (50 pg/mL) required to exacerbate Mtb-induced cell death (Fig. 6B–D and Appendix Fig. S3B and S3D). Thus, it is possible that Mtb-infected MDM-1 secrete low levels of IFN-I but remove it from the medium at nearly the same rate for more than 24 h, so that IFN-I signaling and ISG upregulation occur before IFN-I accumulates in the medium at a level detectable by the assays used here.

## Exogenous IFN-I exacerbates Mtb-induced MDM cell death

Regardless of how much IFN-I Mtb infection induces MDM to secrete, other cells can produce IFN-I in an Mtb-infected host, including conventional dendritic cells (DC) (Giacomini et al, 2001; Remoli et al, 2002) and plasmacytoid DCs (Kotov et al, 2023; Lee et al, 2023). Mimicking the provision of IFN-I by other cells, addition of exogenous IFN-β or IFN-α 4 h post infection led in a dose dependent manner to greater death at 2 days post infection of Mtb-infected MDM-1 (Fig. 6A,B; Appendix Figs. S3A,B and S4A,B). Moreover, priming of MDM-1 with IFN-β or IFN-α 1 day pre-infection led to even greater Mtb-induced cell death (Fig. 6C–E; Appendix Figs. S3C,D and S4C,D; Fig. EV5A). Pre-infection exposure to IFN-β also increased the extent of Mtb-induced cell death in MDM-2 (Fig. EV5B). Without Mtb infection, IFN-I caused no detectable cell death in MDM. This result mirrors what was observed with isolated, Mtb-infected mouse macrophages, whose death was co-dependent on IFN-I induced by Mtb infection and on some additional response to Mtb (Zhang et al, 2020).

## Discussion

This study demonstrates that small amounts of exogenous IFN-I added before or after Mtb infection promote Mtb-induced death of primary human macrophages differentiated in vitro from mono-cytes under two distinct methods, which we have called MDM-1 (Novikov et al (2011)) and MDM-2 (Vogt and Nathan (2011)). Whether IFN-I produced by the Mtb-infected MDM themselves promoted their own death proved to be a more complicated question, in two respects. First, the literature left considerable doubt as to whether Mtb-infected human MDM actually secrete IFN-I. Second, anti-IFN-I antibodies inhibited Mtb-induced ISG expression but did not inhibit Mtb-induced cell death. Below, we discuss each of these issues in turn.

Mtb infection of mouse macrophages activates STING (Collins et al, 2015; Watson et al, 2015), and the human MDM studied here secreted readily detectable IFN-I in response to a chemical compound that activates STING. Human MDM responded to Mtb by secreting TNFα at a time when no IFN-I was detectable in their overlying medium without the STING activator, and they continued to secrete IFN-I in response to the STING activator when they had been infected with Mtb. Yet IFN-I was only detectable in the medium of Mtb-infected MDM at levels >5 pg/mL by 48 h post infection, while ISG upregulation inhibitable by anti-IFN-I antibodies was evident more than 36 h earlier. We suspected that this might be explained by nearly equal rates of IFN-I secretion and depletion from the medium over the first 24 or more hours post infection. Indeed, MDM-1 rapidly depleted exogenous IFN-β when it was added in concentrations comparable to those detected 48 h post Mtb infection. We do not know if the depletion of exogenous IFN-β reflected its retention at the cell surface, its internalization, its degradation, or a combination of these. It is also conceivable that in the first two days of Mtb infection, MDM secreted IFN-I into an intracellular compartment, where it bound IFNAR without first being released from the cell. Studies on other cells have suggested that exposure to low levels of IFN-I can prime cells to produce higher levels of IFN-I when infected (Taniguchi and Takaoka, 2001). Perhaps studying isolated CD14+ monocytes as they differentiate in vitro deprives them of priming, resulting in a limited ability of the MDM to secrete IFN-I upon infection by Mtb. In contrast, in Sp140-/- mice with tuberculosis, interstitial pulmonary macrophages produced IFN-I, perhaps after having been primed by IFN-I from plasmacytoid dendritic cells, which were more prolific producers (Kotov et al, 2023).

In the Sp140-/- mice, bystander interstitial macrophages were major producers of and responders to IFN-I (Kotov et al, 2023). Given that not all the MDM may have been infected, it is not certain whether it is Mtb-infected MDM or uninfected, bystander MDM that are producing IFN-I and/or upregulating ISGs. Also unknown is whether ISGs promote cell death in the same

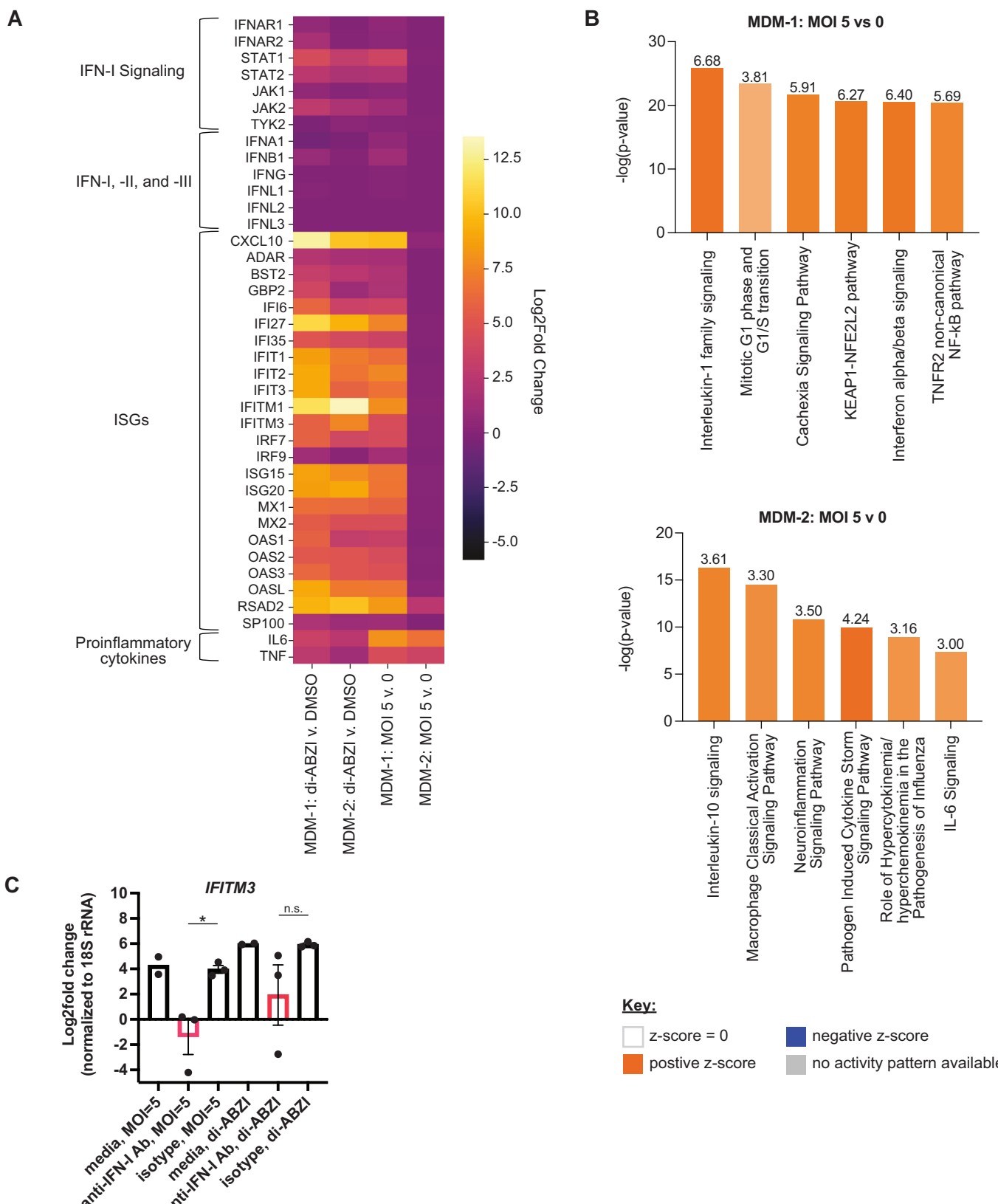

 **Figure 3. Mtb-infected MDM-I upregulate ISGs in an IFN-I dependent manner.**

(A, B) MDM-1 and MDM-2 ($n = 3$) were infected with Mtb H37Rv, treated with 3 µM di-ABZI or vehicle controls (medium or DMSO, final concentration 0.06%). Lysates were collected 1 d post-treatment or post-infection for RNA sequencing. MDM were collected from 3 donors (MDM differentiation and Mtb infection were done with cells from 1–2 donors at a time). (A) Heatmap of $\log_2$ fold changes in gene expression of stimulated/infected MDMs compared to unstimulated MDMs. (B) Ingenuity Pathway Analysis (QIAGEN) of the top 6 canonical pathways upregulated in Mtb-infected MDM-1 (top) and Mtb-infected MDM-2 (bottom) as determined by –log(adjusted $p$-value) and absolute z-scores >3. Numbers above bars display z-scores. (C) Mtb-infected MDM-1 ($n = 2$ or 3) were treated with a 1:1000 dilution anti-IFN-I/anti-IFNAR neutralizing Abs or IgG2a isotype Ab for 2 h pre-infection, infected with Mtb H37Rv, again treated with the respective Ab after Mtb was removed by washing the monolayers 4 h post infection, and lysates were collected 1 d post infection. *IFITM3* expression was measured by RT-qPCR. Data Information: Bar graphs report the mean ± SEM. Each ● represents the average value of 3–4 technical replicates per donor, while $n$ indicates biological replicates. Statistical significance was determined using QIAGEN's IPA analysis, which employs Fisher's exact test and the Benjamini–Hochberg method to correct the FDR (Krämer et al, 2013) (B) and using a one-tailed paired, $t$-test (*$p < 0.5$; n.s. indicates no significance) (C). Source data are available online for this figure.

macrophages in which the ISGs are induced. Further investigation using reporter cells or sorting of uninfected and infected cells could address these questions. We found that IFN-I signaling alone did not induce MDM death; Mtb infection was required. Perhaps Kotov et al (2023) found fewer Mtb infected interstitial macrophages than uninfected interstitial macrophages that were IFN-I producing and IFN-I-responsive in vivo because Mtb-infected interstitial macrophages responsive to IFN-I had already died.

While ISG induction was blocked by anti-IFN-I antibodies, cell death was not. Nonetheless, we suspect that endogenous IFN-I produced in response to Mtb infection may contribute to Mtb-induced death of MDM. Supporting this speculation, MDM-2, which showed little induction of ISGs in response to Mtb infection, were far more resistant to cell death induced by Mtb infection at low MOIs than MDM cultured under the protocol we called MDM-1 or any variation of it tested in the earlier study (Vogt and Nathan, 2011). Perhaps some responses to IFN-I are initiated when IFN-I binds IFNAR at the cell surface, while other responses are initiated when IFN-I binds IFNAR in endocytic vesicles. Anti-IFN-I antibodies that intercept IFN-I at the cell surface might disengage from it within the acidified endosomes, allowing signaling to proceed. Endosomal signaling by IFN-I has been described. For example, human retinal pigment epithelial-1 and HeLa cell lines transport the IFN-I:IFNAR complex to an endosomal compartment, where IFN-I signaling occurs (Chmiest et al, 2016; Zanin et al, 2021; Zanin et al, 2023).

For nearly a hundred years, investigators have studied the biology of human macrophages by differentiating monocytes in vitro (Lewis, 1925). However, there is still no standard method of monocyte culture and MDM differentiation, and few studies have compared different methods. Most often used is the method we have called MDM-1 and variations thereof. The method we have called MDM-2 arose from systematic comparison of media, atmospheres, sources and amounts of serum and plasma, types of cytokines, cytokine combinations, cytokine doses, sequences of cytokine addition, and durations of culture to best equip MDM to preserve their viability after infection by Mtb (Vogt and Nathan, 2011). The present study is the first, to our knowledge, to compare any two culture methods for their impact on IFN-I secretion by MDM and the impact of IFN-I on Mtb-infected MDM. The key features limiting IFN-I induction and ISG expression in Mtb-infected MDM-2 were the use of 40% autologous plasma and the extension of culture to a second week before infection. Further studies will be needed to determine if the differential impact of 10% FBS versus 40% human plasma is due to the amounts of these materials, the species from which they are derived, the depletion of

coagulation factors upon formation of serum, or the heat-inactivation of FBS, which inactivates complement, among other variables.

In conclusion, it remains unsettled whether low levels of *endogenously produced* IFN-I contribute to the death of Mtb-infected MDM in vitro. Nonetheless, as little as 50 pg/mL of *exogenous* IFN-I can do so, and such concentrations are likely to be present at sites of the disease (Kotov et al, 2023). Death of human MDM that is co-dependent on Mtb and on IFN-I is expected to allow growth of Mtb in necrotic MDM and facilitate the spread of Mtb within the host (Lerner et al, 2017; Mahamed et al, 2017; Toniolo et al, 2023). This may be one mechanism by which IFN-I exacerbates TB. Blocking IFN-I itself would expose a host to considerable risk from viral infections. However, identification of a step downstream in the IFN-I signaling pathway that leads to cell death without conferring antiviral protection could open an avenue to host-directed therapy of TB, as had been proposed based on studies in mice (Dorhoi et al, 2014; Zhang et al, 2020).

## Methods

### Reagents and tools table

| Reagent/Resource | Reference or Source | Identifier or Catalog Number |
|---|---|---|
| **Experimental Models** | | |
| Monocyte-derived macrophages | Human blood obtained in this study | |
| HEK-Blue™ IFN-α/β cells (and corresponding reagents) | InvivoGen | Cat # hkb-ifnabv2 |
| HEK-Blue™ IFN-λ Cells (and corresponding reagents) | InvivoGen | Cat # hkb-ifnlv2 |
| U937 | ATCC | Cat # CRL-1593.2 |
| THP-1 | ATCC | Cat # TIB-202 |
| *Mycobacterium tuberculosis* strain H37Rv | ATCC | Cat # 25618 |
| *Mycobacterium tuberculosis* strain H37Rv pG13-tdTomato | From William R. Jacobs, Jr's lab | |
| *Mycobacterium tuberculosis* clinical strain Mtb-LT1 | (Verma et al, 2019) | |
| *Mycobacterium tuberculosis* clinical strain Mtb-HT1 | (Verma et al, 2019) | |
| **Cytokines** | | |
| Recombinant human M-CSF | R&D Systems | Cat # 216-MC-010 |
| Recombinant human GM-CSF | R&D Systems | Cat # 215-GM-050 |
| Recombinant human IFN-α 2a | PBL Assay Science | Cat # 11100-1 |

| Reagent/Resource | Reference or Source | Identifier or Catalog Number |
|---|---|---|
| Recombinant human IFN-β | PBL Assay Science | Cat # 11415-1 |
| Recombinant human IFN-γ | Roche; Millipore Sigma | Cat # 11040596001 |
| Recombinant human IFN-λ1 | Peperotech | Cat # 300-02 L |
| Recombinant human TNFα | Biolegend | Cat # 570102 |
| **Antibodies** | | |
| Human IFN-I neutralzing antibody mixture, pAb | PBL Assay Science | Cat # 39000-1 |
| Mouse IgG2aκ | BD Pharmingen | Cat # 555574 |
| Human CD14 MicroBeads, from mouse | Macs Miltenyi Biotec | Cat # 130-050-201 |
| Human anti-IFNAR1, pAb, from goat | Abcam | Cat # ab10739 |
| Human anti-IFNAR2, mAb, from mouse | PBL Assay Science | Cat # 21385-1 |
| Human pSTAT1 (Y701), from rabbit | Cell Signaling | Cat # 7649 S |
| Human/mouse STAT1, from rabbit | Cell Signaling | Cat # 9172 |
| Human/mouse β-actin | Santa Cruz | Cat # sc-47778 |
| Anti-mouse, horseradish peroxidase (HRP) linked antibody | Cell Signaling | Cat # 7076 |
| Anti-rabbit, HRP linked secondary antibody | Cell Signaling | Cat # 7074 |
| **Oligonucleotides and sequence-based reagents** | | |
| ISG15 forward primer: GAGAGGCAGCGAACTCATC | TIBMol | |
| ISG15 reverse primer: CAGGGACACCTGGAATTCGTT | TIBMol | |
| ISG15 forward probe: TGCCAGTACAGGAGCTTGTGCC, LC500 5' modification and a BlackBerry Quencher (BBQ) 3' modification | TIBMol | |
| IFITM3 forward primer: CTGCTGCCTGGGCTTCATAG | TIBMol | |
| IFITM3 reverse primer: CGTCGCCAACCATCTTCCT | TIBMol | |
| IFITM3 forward probe: TCGCCTACTCCGTGAAGTCTAGGG with LC610 5' modification and a Black Hole Quencher 2 (BHQ2) 3' modification | TIBMol | |
| 18S forward primer: GCATGGCCGTTCTTAGTTGG | TIBMol | |
| 18S reverse primer: TGCCAGAGTCTCGTTCGTTA | TIBMol | |
| 18S forward probe: TGGAGCGATTTGTCTGGTTAATTCCGA with a LC670 5' modification and a BBQ 3' modification | TIBMol | |
| **Chemicals, enzymes, and other reagents** | | |
| Heparinized tubes | BD Vacutainer | Cat # 367874 |
| RPMI-1640 | Gibco | Cat # 11875-093 |
| Dulbecco's phosphate-buffered saline (PBS) | Gibco | Cat # 14190-144 |
| GlutaMAX | Gibco | Cat # 350501-061 |
| 10 mM HEPES | Gibco | Cat # 15630-080 |
| 1 mM sodium pyruvate | Gibco | Cat # 11360-070 |
| L-glutamine 200 mM (100X) | Gibco | Cat # 25030-081 |
| FBS | Cytiva | Cat # SH30070.03 |
| Ethylenediaminetetraacetic acid (EDTA) | Invitrogen | Cat # 15575-038 |
| Pen Strep | Gibco | Cat # 15140-122 |

| Reagent/Resource | Reference or Source | Identifier or Catalog Number |
|---|---|---|
| Ficoll-Paque Plus | Cytiva | Cat # GE17-1440-02 |
| Di-amidobenzimidazole (di-ABZI) | MedChemExpress | Cat # HY-112921A |
| Difco Middlebrook 7H9 broth | BD Bioscience | Cat # 271310 |
| Glycerol (for molecular biology) | Sigma-Aldrich | Cat # G5516 |
| Oleic acid-dextrose-catalase (OADC) | BD Biosciences | Cat # 212351 |
| Tyloxapol | Sigma-Aldrich | Cat # 25301-02-4 |
| Beta Mercapatolethanol | Gibco | Cat # 31350-010 |
| High sensitivity IFNb ELISA | PBL Assay Science | Cat # 41415 |
| TNFα ELISA | Abcam | Cat # ab181421 |
| CellTiter-Glo Luminescent Cell Viability Assay | Promega | Cat # G7573 |
| Spin-X® Centrifuge Tube Filters, 0.22 μm Pore CA Membrane, Sterile | Costar | Cat # 8160 |
| RNeasy Mini kit | Qiagen | Cat # 74106 |
| Rnase-free Dnase set | Qiagen | Cat # 79256 |
| Qubit RNA High Sensitivity Range Assay kit | Invitrogen | Cat # Q32852 |
| SuperScript III First-Strand Synthesis System | Invitrogen | Cat # 18080-093 |
| SuperScript IV First-Strand Synthesis System | Invitrogen | Cat # 18090050 |
| 10 mM dNTP Mix | Invitrogen | Cat # 18427-013 |
| oligo dT(20) primers | Invitrogen | Cat # 18418020 |
| RNaseOUT Ribonuclease Inhibitor | Invitrogen | Cat # 10777-019 |
| RNAse H | Invitrogen | Cat # 18021-014 |
| LightCycler 480 Probes Master | Roche | Cat # 4707494001 |
| Trizol Reagent | Ambion by Life Technologies | Cat # 15596018 |
| Chloroform:isoamyl alcohol (24:1) | Sigma | Cat # C0549-1PT |
| DEPC treated water | Invitrogen | AM9915G |
| Kanamycin | Sigma-Aldrich | Cat # K4000 |
| 7-amino-actinomycin D (7-AAD) | BD Bioscience | Cat # 559925 |
| Phorbol 12-myristate 13-acetate (PMA) | Sigma-Aldrich | Cat # P8139 |
| Sodium dodecyl-sulfate (SDS) | Sigma-Aldrich | Cat # L3771 |
| Glycerol | Sigma | Cat # G7893 |
| Trisma base (Tris base) | Sigma-Aldrich | Cat # T1503 |
| Tris hydrochloric acid (HCL) | JT Baker | Cat # 4103-06 |
| HCL | JT Baker | Cat # 5619-02 |
| Bromophenol blue | Sigma-Aldrich | Cat # B0126 |
| Dithithretol (DTT) | Denville Scientific | Cat # CD-4070-4 |
| 30% (w/v) acrylamide: 0.8% (w/v) bis-acrylamide | National Diagnostics | Cat # EC-890 |
| Ammonium persulfate (AP) | Sigma | Cat # A9164 |
| N,N,N',N'-Tetramethylethylenediamine (TEMED) | Sigma | Cat # T9281 |
| Methanol | JT Baker | Cat # 9070-05 |
| Glycine | BioRad | Cat # 1610724 |
| Polyvinylidene difluoride (PVDF) membranes, 0.45 μm pore size | Millipore | Cat # IPVH00010 |

| Reagent/Resource | Reference or Source | Identifier or Catalog Number |
|---|---|---|
| Blotting paper Grade 707 | Avantor | Cat # 28298 |
| Tween 20 | Sigma | Cat # P1379 |
| Sodium chloride (NaCl) | JT Baker | Cat # 3624-07 |
| Amersham ECL Prime Western Blotting Detection Reagent | Cytiva | Cat # RPM2232 |
| Clarity and Clarity Max ECL Western Blotting Substrates | BioRad | Cat # 10026385 |
| X-ray film | Denville Scientific | Cat # XC59X |
| Restore™ PLUS Western Blot Stripping Buffer | Thermo Scientific | Cat # 46430 |
| **Software** | | |
| GraphPad Prism v10 | http://www.graphpad.com | |
| FIJI | https://fiji.sc/ | |
| Python (Anaconda) | https://www.anaconda.com/ | |
| QIAGEN Ingenuity Pathway Analysis | https://digitalinsights.qiagen.com/products-overview/discovery-insights-portfolio/analysis-and-visualization/qiagen-ipa/ | |
| Interferome | https://interferome.org/interferome/home.jspx | |
| **Other** | | |
| NovaSeq 6000 | Illumina | |
| LightCycler 480 Instrument | Roche | |
| NanoDrop OneC Microvolume UV-Vis Spectrophotometer | Thermo Scientific | |
| Qubit 3 Fluorometer | Invitrogen | |
| Agilent 2100 Bioanalyzer | Agilent | |
| Keyence BZ-X800 fluorescent microscope | Keyence | |
| SpectraMax iD5 | Molecular Devices | |
| SpectraMax® L Luminescence Microplate Readers | MDS Analytical Technologies | |
| SpectraMax M5 | Molecular Devices | |
| Genesys 20 spectrometer | Thermo Scientific | |
| Semi-dry electrophoretic transfer cell | Bio-Rad | |
| Autoradiography cassette | Cytiva | RPN11648 |
| Film Processor | MXR Imaging | |

## Cell culture

16 total donors were healthy adult men and women who reflected the demographic diversity of New York City and provided written informed consent, which abides by the principles stated in the WMA Declaration of Helsinki and the Department of Health and Human Services Belmont Report. Blood was collected by venipuncture into heparinized tubes (BD Vacutainer). For MDM-2, we followed the "optimized method" described in Vogt and Nathan (2011). In brief, blood was mixed with an equal volume of RPMI-1640 (GIBCO, Invitrogen), supplemented with 2 mM GlutaMAX (Invitrogen), and centrifuged above Ficoll-Paque Plus (GE Healthcare) at $500 \times g$ for 20 min at 20–22 °C. The upper plasma layer was collected, centrifuged at $2000 \times g$ for 2 h at 20–22 °C to remove platelets and the final supernatant used to supply 40% autologous plasma, resulting in "complete medium 2". The peripheral blood mononuclear cells (PBMC) were collected, and the monocytes isolated by CD14 magnetic bead selection (MACS; Miltenyi Biotec) as described in Vogt and Nathan (2011).

Monocytes were stimulated with 0.5 ng/mL each of recombinant human GM-CSF (R&D Systems) and TNFα (Biolegend). The two cytokines were replenished every 3–4 days for 2 weeks by replacing 30% of complete medium 2 containing 3 times the final concentration of cytokines. For MDM-1, because it is unclear whether Novikov et al (2011) purified CD14+ monocytes, the Vogt and Nathan (2011) method of isolating monocytes was used so that the monocytes purified would be similar between the two methods. However, for isolation of PBMC for differentiation of MDM-1, the blood was mixed with an equal volume of complete medium 1: RPMI-1640 (GIBCO, Invitrogen), 10% FBS (Cytiva), 2 mM glutamate (Gibco), 10 mM HEPES buffer (Gibco), and 50 µM β-mercaptoethanol (BME) (Gibco), as described in Novikov et al (2011). Monocytes were then cultured in medium 1 and stimulated with 10 ng/mL of recombinant human M-CSF (R&D Systems), which was replenished on days 2 and 4 post isolation for 7 days by replacing 30% of complete medium 2 containing 3 times final concentration of M-CSF. Isolated monocytes were seeded into tissue culture treated plates at a concentration of 100,000 cells/0.2 mL. White-walled 96-well plates were used for cell viability assays, clear walled 96-well plates for cytokine measurements, 12-well plates for RNA sequencing and/or 24-well plates for real time-quantitative polymerase chain reaction (RT-qPCR).

THP-1 (ATCC) and U937 (ATCC) were cultured in RPMI-1640 (GIBCO, Invitrogen), supplemented with 10% FBS (Cytiva), 2mM L-glutamine (Gibco), 10 mM HEPES (Gibco), and 1 mM sodium pyruvate (Gibco). THP-1 was also supplemented with 50 µM BME. Both cell lines tested negative for mycoplasma by Hoechst stain. THP-1 and U937 were stimulated with 10 ng/mL of PMA for 2 days prior to Mtb H37Rv infection and then infected with Mtb as described for MDM.

## Mtb culture and infection of macrophages

Mtb H37Rv was used for most experiments. Where indicated, we used recent clinical isolates, low transmission stain 1 (clinical strain 1) and high transmission strain 1 (clinical strain 2), generously shared by Christopher Brown (Verma et al, 2019). When visualizing Mtb, we used a strain of H37Rv transformed with a plasmid expressing the red fluorescent protein tdTomato and a kanamycin resistance gene under a *M. marinum* G13 promoter (pYUB1169 or pG13-tdTomato (a generous gift from Travis Hartman, who prepared it while in the lab of W. R. Jacobs, Jr.). Mtb was grown to log phase in Middlebrook 7H9 broth (BD Difco) supplemented with 0.5% glycerol (Sigma-Aldrich), 10% oleic acid-dextrose-catalase (OADC) (BD Biosciences) and 0.02% tyloxapol (Sigma-Aldrich) at 37 °C and 5% $CO_2$. Mtb was cultured in antibiotic-free medium, except for tdTomato-Mtb, which was cultured with 40 µg/mL kanamycin (Sigma-Aldrich, K4000). Kanamycin was washed out when single-cell suspensions of Mtb were prepared to infect MDMs. Single-cell suspensions were prepared by pipetting pelleted Mtb through a 1 mL micropipette 30 times and centrifuging the suspension for 10 min at $200 \times g$ with no brake, as detailed in Zhang et al (2020). Based on determination that 0.1 $OD_{580}$ of the non-sedimented suspension = $5 \times 10^7$ bacteria/mL, the requisite amount of Mtb was centrifuged at $3082 \times g$ for 8 min, resuspended in the medium used for MDM-1 or MDM-2, and added to MDMs. 4 h post Mtb infection, extracellular

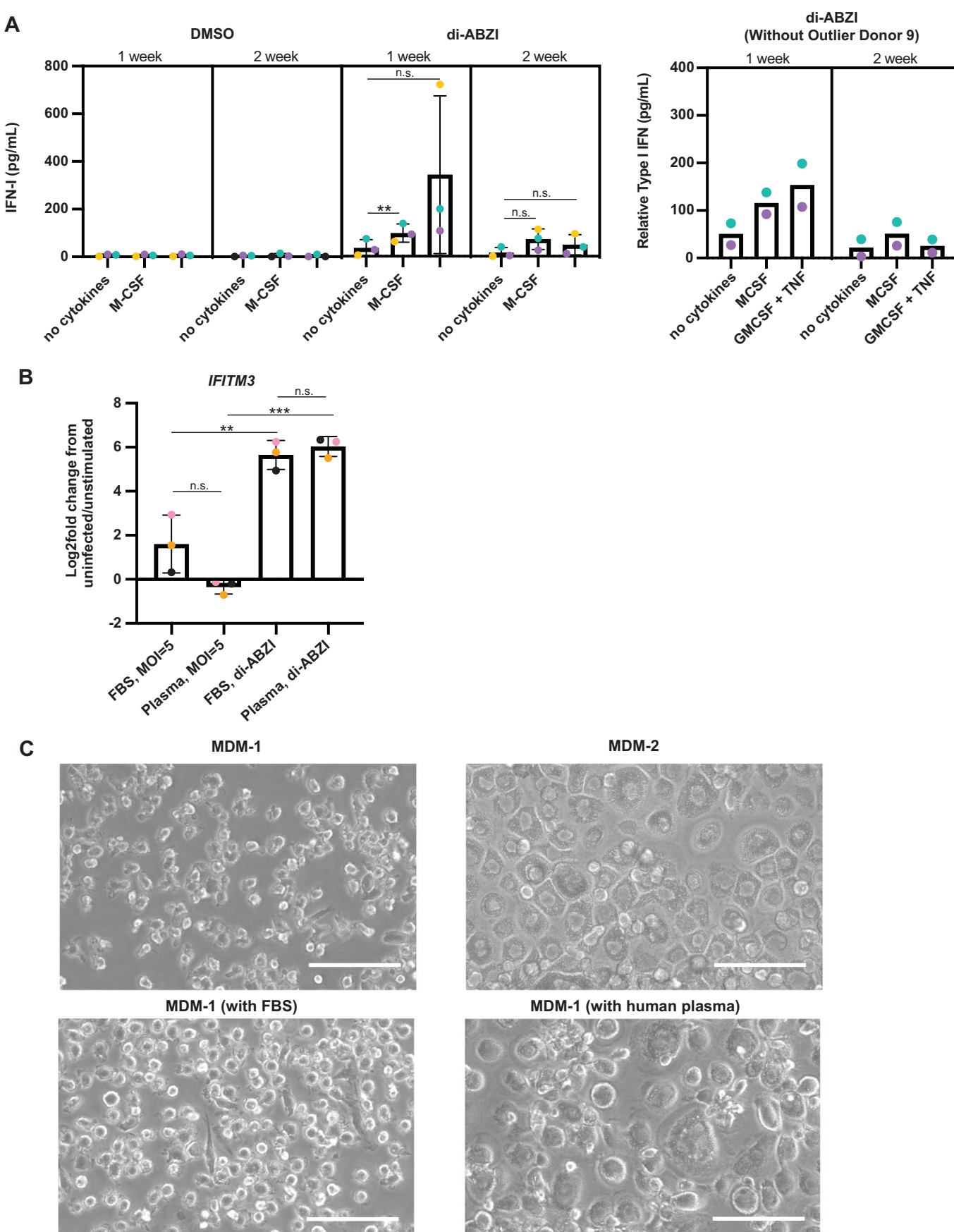

◄ **Figure 4. Length of differentiation time and type of serum or plasma in the medium determine the level of IFN-I secretion upon STING activation and the level of ISG upregulation upon Mtb infection of MDMs.**

(**A**) MDMs ($n = 3$) were differentiated under method 2, except that they were cultured without cytokines, with 10 ng/mL M-CSF, or with 0.5 ng/mL GM-CSF and TNFα for 1 or 2 weeks. MDMs were stimulated with 3 μM di-ABZI and supernatants were collected 1 d post infection. IFN-I expression in the supernatant was measured via the IFN-I HEKBlue reporter assay and quantified by performing a sigmoidal, 4 parameter interpolation from an IFN-β standard curve. Right figure displays all 3 donors. Left image displays the MDMs of 2 donors without the outlier. (**B, C**) MDMs ($n = 3$) were differentiated under method 1, except that they were cultured with 10% FBS or 40% autologous human plasma and infected with Mtb H37Rv. Lysates were collected 1 d post infection. Values too low to be interpolated were assigned a value of 1 pg/mL. (**B**) *IFITM3* expression was measured by RT-qPCR. (**C**) Representative images of MDM-1 (upper, left) ($n = 3$) 9 days post differentiation, or MDM-2 (upper, right) ($n = 3$) 16 days post differentiation isolated from different donors. Representative images ($n = 3$) of MDM-1 from the same donor differentiated under method 1 (bottom, left) or MDM-1 differentiated under method 1 except in 40% autologous human plasma instead of FBS (bottom, right) 7 days after differentiation. Scale bars = 100 μm. Images were taken at 20X using phase contrast. Data Information: Bar graphs report the mean ± SEM. Each ● represents the average value of (**A**) 3 technical replicates per donor or (**B**) 3–4 technical replicates per donor. (**A, B**) Statistical significance was determined using a two-tailed paired, *t*-test (**$p < 0.01$; ***$p < 0.001$; n.s. indicates no significance). In each panel, *n* indicates biological replicates. Source data are available online for this figure.

Mtb was washed out twice with PBS pre-warmed to 37 °C. MDM medium pre-warmed to 37 °C was added again.

## Cell viability

Cell viability was determined by measuring cellular ATP levels with the CellTiter-Glo kit (Promega) according to the manufacturer's instructions, with the exception that 30% of reagent volume was used. For the MDMs treated with antibodies, the percentage of cell viability was normalized to the values for uninfected cells. For the MDMs that were treated with IFN-α, IFN-β, or IFN-γ, the percentage of cell viability was normalized to values for cells that were neither infected nor IFN-treated. Given the basal heterogeneity in ATP levels of MDMs from different donors, the cell viability was normalized to each donor's uninfected, non-primed MDMs to better reveal the effects of IFN-I or its inhibition on the MDMs.

## Cytokine measurements

At indicated timepoints, supernatants of cells infected with Mtb or stimulated with di-amidobenzimidazole (di-ABZI) (MedChemExpress) were collected and centrifuged at $21,130 \times g$ for 5 min at 4 °C. The supernatants were filtered through a low-protein retention, 0.22 μm cellulose acetate filter by centrifugation at $6010 \times g$ for 2 min at 4 °C, removed from the biosafety level 3 facility and stored at −80 °C. All IFN-I subtypes were measured using IFN-α/β Reporter HEK 293 T cells (InvivoGen) according to the manufacturer's instructions. The HEK 293T cells were guaranteed to be mycoplasma free by the manufacturer and confirmed to be negative for mycoplasma by Hoechst stain. The concentration of IFN-I was determined by performing a sigmoidal, 4 parameter interpolation from an IFN-β standard curve, using Prism's software. Absorbance values below the standard curve that were too low to be interpolated were assigned a value of 1 pg/mL of IFN-I, and values too high to be interpolated were assigned a value of 4800 pg/mL of IFN-I. TNFα was measured using an ELISA (Abcam) according to the manufacturer's instructions. Absorbance values below the standard curve that were too low to be interpolated were assigned a value of 1 pg/mL of TNFα, and values too high to be interpolated were assigned a value of 2000 pg/mL of TNFα. IFN-β was measured using a high-sensitivity ELISA (PBL Assay Science, 41415) according to the manufacturer's instructions. IFN-III was measured using an IFN-l Reporter HEK 293 cells (InvivoGen), according to

the manufacturer's instructions. The concentration of IFN-III was determined by performing a sigmoidal, 4 parameter interpolation from an IFN-λ1 (IL-29) (Peprotech, 300-02L) standard curve, using Prism's software.

## RNA sequencing and qPCR

At indicated timepoints, MDMs were washed once with PBS and homogenized with 1 mL TRIzol Reagent (Invitrogen) per $1 \times 10^6$ cells. 200 μL chloroform:isoamyl alcohol (24:1) (Sigma-Aldrich) was added per 1 mL TRIzol reagent. The mixture was shaken vigorously for 15 s and then centrifuged at $12,000 \times g$ for 15 min at 4 °C. The upper aqueous layer containing RNA was removed from the biosafety level 3 facility and stored at −80 °C. RNA was further purified using a RNeasy Mini kit (Qiagen), including the steps for "On column DNase digestion" (Qiagen). RNA concentration was measured using a NanoDrop One^C Microvolume UV-Vis Spectrophotometer (Thermo Scientific) or a Qubit 3 Fluorometer (Invitrogen) using the Qubit RNA High Sensitivity Range Assay kit (Invitrogen) according to the manufacturer's instructions. RNA quality was measured using an Agilent 2100 Bioanalyzer in the Weill Cornell Medicine Genomics (WCM) Resource Core Facility, where RNA with an integrity number >7 was sequenced with paired-end 50 base pairs on a NovaSeq 6000 instrument. To identify ISGs, RNA sequence data sets were run through an IFN regulated gene database, Interferome (Rusinova et al, 2012).

The WCM Resource Core Facility converted raw RNA sequencing reads in BCL format to FASTQ files and demultiplexed them using bcl2fastq 2.20 (Illumina). The adaptors were trimmed using cutadapt (version1.18) (https://cutadapt.readthedocs.io/en/v1.18/). RNA reads were aligned and mapped to the GRCh38 human reference genome using STAR (Version 2.5.2) (https://github.com/alexdobin/STAR) (Dobin et al, 2013). The transcriptome reconstruction was performed via Cufflinks (Version 2.1.1) (http://cole-trapnell-lab.github.io/cufflinks/), and the transcript abundance was measured with Cufflinks in Fragments Per Kilobase of exon model per Million mapped reads (FPKM) (Trapnell et al, 2013; Trapnell et al, 2010). HTSeq-count v0.11.2 was used to extract the raw read counts per gene (Anders et al, 2015). Differential gene expression was analyzed using the DESeq2 package (https://bioconductor.org/packages/release/bioc/html/DESeq2.html) (Love et al, 2014).

For real-time quantitative PCR (RT-qPCR), cDNA was synthesized from RNA using SuperScript III or IV First-Strand Synthesis System

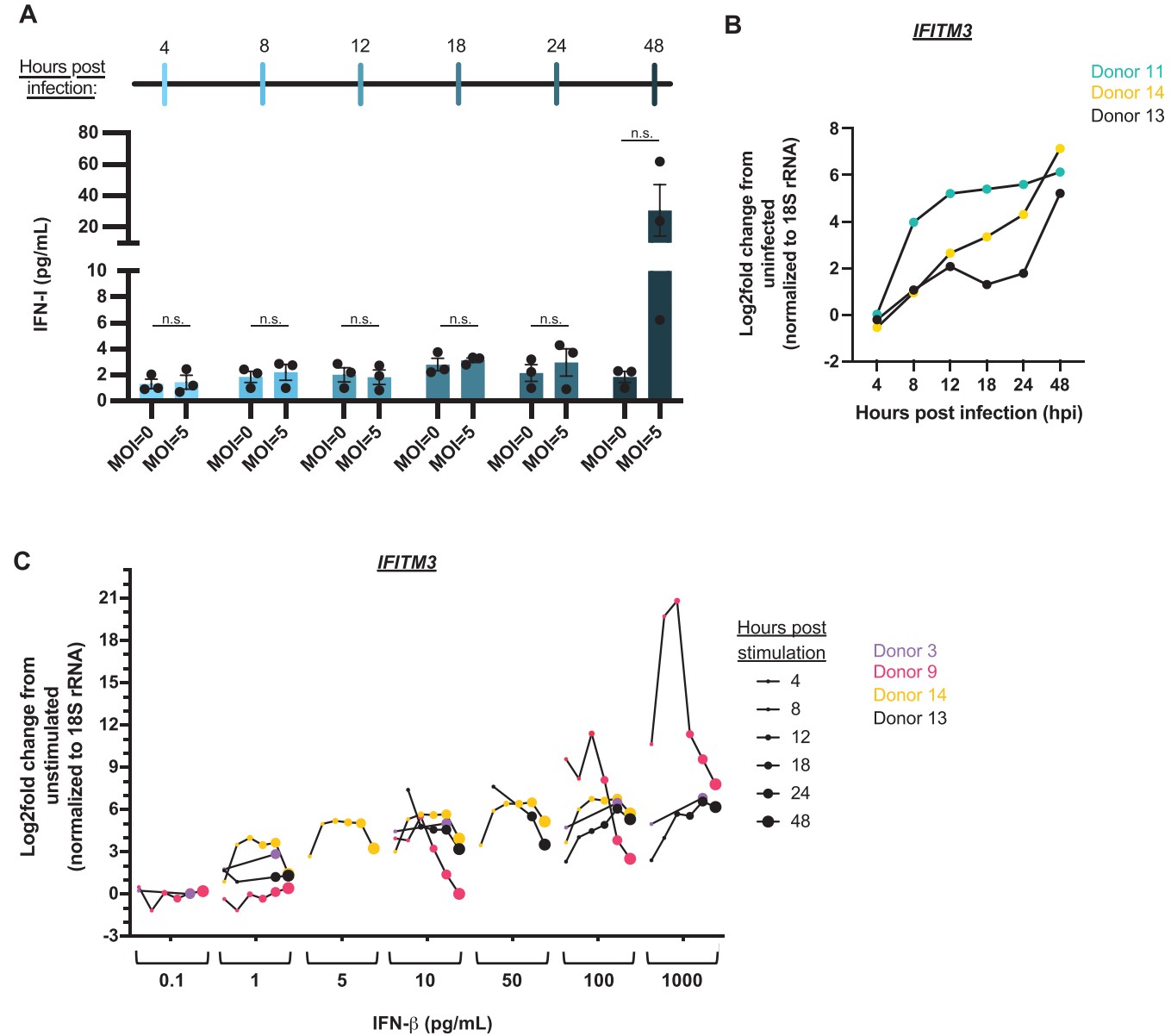

**Figure 5. Mtb-infected MDM-1 upregulated ISGs earlier than they secreted IFN-I.**

MDM-1 were infected with Mtb H37Rv and supernatants and lysates were collected at indicated timepoints. For the timepoint 4 h post infection, extracellular Mtb was not washed out. (A) IFN-I expression was measured via the IFN-I HEKBlue reporter assay and quantified by performing a sigmoidal, 4 parameter interpolation from an IFN-β standard curve. Values too low to be interpolated were assigned a value of 1 pg/mL. (B) IFITM3 expression was measured by RT-qPCR. (C) MDM-1 ($n = 1$–3) were stimulated with 0.1–1000 pg/mL IFN-β, lysates were collected at indicated timepoints, and IFITM3 expression was measured by RT-qPCR. Data Information: Bar graph reports the mean ± standard error of mean (SEM) ($n = 3$). Each ● represents the average value of 2 technical replicates (A) or 3–4 technical replicates (B and C) per donor, $n$ indicates biological replicates. (A) Statistical significance was determined using a two-tailed paired, $t$-test (n.s. indicates no significance). (B) Correlation between time post infection and log2fold change of IFITM3 from uninfected was determined by simple linear regression of the means of the donors per timepoint (R squared = 0.8282). Source data are available online for this figure.

(Invitrogen), oligo dT(20) primers (Invitrogen), and RNAse H (Invitrogen) post cDNA synthesis, according to the manufacturer's instructions. RT-qPCR was performed using LightCycler 480 Probes Master (Roche) and read by LightCycler 480 Instrument (Roche), according to the manufacturer's instructions. ISG15 was detected using GAGAGGCAGCGAACTCATC (forward), CAGGGACACCTGGAATTCGTT (reverse), and TGCCAGTACAGGAGCTTGTGCC probe with a LC500 5' modification and a BlackBerry Quencher (BBQ) 3' modification made

by TIBMol. 18S RNA was detected using GCATGGCCGTTCTTAG TTGG (forward), TGCCAGAGTCTCGTTCGTTA (reverse), and TGGAGCGATTTGTCTGGTTAATTCCGA probe with a LC670 5' modification and a BBQ 3' modification made by TIBMol. IFITM3 was detected using CTGCTGCCTGGGCTTCATAG (forward), CGTCG CCAACCATCTTCCT (reverse), and TCGCCTACTCCGTGAAGTCTA GGG probe with LC610 5' modification and a Black Hole Quencher 2 (BHQ2) 3' modification made by TIBMol. For any gene in a sample in

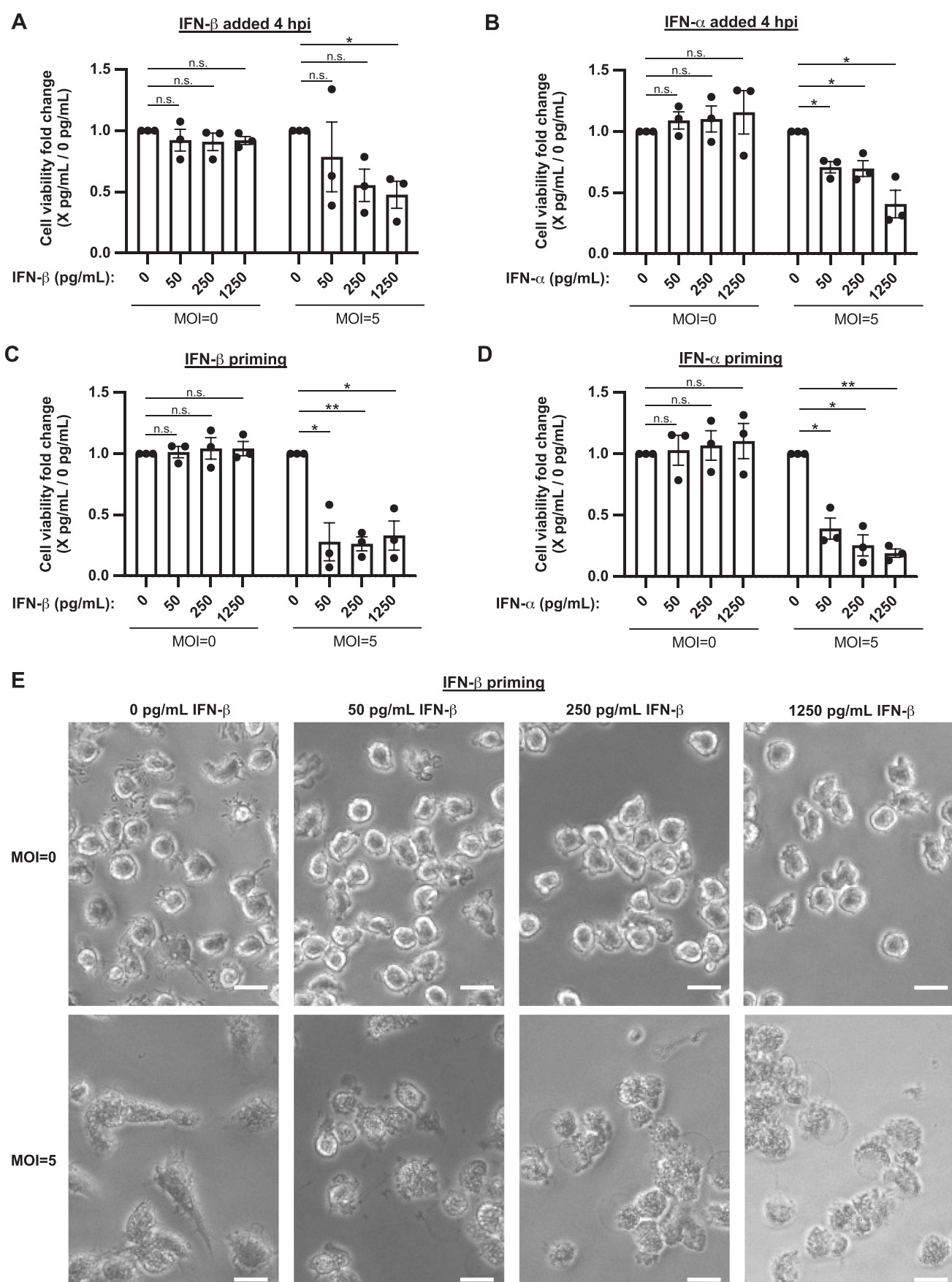

**Figure 6. Exogenous IFN-I exacerbates Mtb-induced death of MDM-1.**

(A) IFN-β or (B) IFN-α were added to Mtb H37Rv infected MDM-1 ($n = 3$) at 4 h post infection after extracellular Mtb was washed out. Cell viability was determined by CellTiter Glo assay at 2 d post infection. MDM-1 were primed with (C) IFN-β or (D) IFN-α; 1 day later MDM-1 ($n = 3$) were infected with Mtb H37Rv. Cell viability was measured by CellTiter Glo assay at 2 d post infection. Cell viability was normalized to values for the respective uninfected, unstimulated or non-primed MDMs. The fold change of the normalized cell viability was calculated by dividing the normalized cell viability of IFN-I stimulated MDMs by that of unstimulated MDMs within the same group (uninfected or infected) per donor. (E) Images of MDM-1 ($n = 3$) primed with IFN-β 1 day pre-infection, infected with Mtb H37Rv, and imaged at 20x at 2 d post infection. Scale bars = 20 μm. Data Information: (A–D) Bar graphs report the mean ± SEM of the fold change. Each ● represents the average value of 3 technical replicates per donor ($n = 3$), $n$ indicates biological replicates. Cells from each donor were tested in independent experiments. Statistical significance was determined using a paired, two-tailed $t$-test (*$p < 0.5$; **$p < 0.01$; n.s. indicates no significance). Source data are available online for this figure.

which the cycle threshold (Ct) value was not provided, a Ct value of 40 was assigned. Log$_2$fold change was calculated by taking the $2^{-\Delta\Delta Ct}$. $\Delta\Delta Ct =$ (Ct$_{experimental\ gene\ of\ stimulated\ MDM}$ − Ct$_{houskeeping\ gene\ of\ stimulated\ MDM}$) − (Ct$_{experimental\ gene\ of\ unstimulated\ MDM}$ − Ct$_{houskeeping\ gene\ of\ unstimulated\ MDM}$).

## Microscopy

Images of cells and bacteria were recorded using a Keyence BZ-X800 fluorescent microscope and camera. MDM and tdTomato-Mtb were imaged using the ×20 objective (Keyence, BZ-PF20LP) with a 0.45 numerical aperture. MDM were visualized using phase contrast and tdTomato Mtb was visualized using Cy3/TRITC filter cube (Keyence, model OP-87764).

To visualize cell death, Mtb-infected MDM were stained with 7-amino-actinomycin D (7-AAD) (BD Bioscience, 559925), according to the manufacturer's instructions. MDM were imaged using the ×10 objective (Keyence, BZ-PA10) with a 0.45 numerical aperture. MDM were visualized using brightfield and 7-AAD stain was visualized using the Cy5 filter cube (Keyence, model OP-87766).

## Western blot

Lysates were prepared by washing MDM with PBS and lysing MDM with sodium dodecyl-sulfate (SDS) loading buffer: 2% (w/v) SDS, 10% (v/v) glycerol, 0.05 M Tris-hydrochloric acid (HCL) (pH 6.8), 0.1% (w/v) bromophenol blue and 100 mM dithiothreitol (DTT). Lysates were transferred to microcentrifuge tubes, heated at 95–100 °C for 30 min, centrifuged for 2 min at 16,363 × $g$, vortexed at highest speed to mix, and collected by centrifuging for 1 min at 16,363 × $g$. To run SDS polyacrylamide gel electrophoresis (SDS-PAGE), lysates were loaded onto polyacrylamide gel: 15% acrylamide (bottom)—15%/0.4% (w/v) acrylamide/bis-acrylamide, 0.39 M Tris (pH 8.8), 0.1% (w/v) SDS, 0.1% (w/v) ammonium persulfate (AP), and 0.1% (v/v) tetramethylethylenediamine (TEMED); 2% acrylamide (top)—2%/0.05% (w/v) acrylamide/bis-acrylamide, 0.125 M Tris-HCL (pH 6.8), 0.1% (w/v) SDS, 0.1% (w/v) AP, 0.1% (v/v) TEMED. SDS-PAGE was run at with amps per gel constant at 0.02 for the 1st 10 min and then raised to 0.03 until bromophenol blue reached the bottom of gel.

Proteins were transferred to polyvinylidene difluoride (PVDF) membranes with a 0.45 μm pore size via semi-dry transfer (Millipore, IPVH00010). Prior to semi-dry transfer run, PVDF membranes were rinsed with 100% methanol and semi-dry transfer buffer: 5.82 g of Tris-base, 2.93 g glycine, 0.375 g SDS, and 200 mL methanol in 1 L of water. Blotting paper (Avantor, 28298) was soaked with semi-dry transfer buffer. Semi-dry transfer was run on a semi-dry electrophoretic transfer cell (Bio-Rad) with voltage set

as constant at 24 volts and 0.4 amps for 12 min per 1 mm gel. PVDF membrane was blocked for at least 1 h with 5% (w/v) milk in Tris Buffered Saline with Tween 20 (TBST). TBST: 150 mM sodium chloride (NaCl), 20 mM Tris-HCL, and 0.1% Tween 20. PVDF was washed 1 time before rocked with 1:1000 STAT1 antibody (Cell Signaling, 9172S), pSTAT1 antibody (Cell Signaling, 7649S), or b-actin antibody (Santa Cruz, sc-47778) in TBST for at least 2 h or for 1 h for b-actin Ab. After washing 3 times with TBST, PVFD membrane was rocked with respective 1:5000 anti-rabbit, horse-radish peroxidase (HRP) linked secondary antibody (Cell Signaling, 7074) or 1:10,000 anti-mouse, HRP linked antibody (Cell Signaling, 7076) in 5% milk in TBST for 1 h. PVDF membranes were washed 3 times in TBST.

STAT1 and pSTAT1 were visualized by incubating with Amersham ECL Prime Western Blotting Detection Reagent (Cytiva, RPM2232), consisting of 1:1 luminol solution: peroxide solution for 4 min. PVDF membranes were transferred to an X-ray cassette, and a sheet of X-ray film (Denville Scientific, XC59X) was placed on top in a dark room. X-ray film was developed with a film processor (MXR Source One).

## Statistics

Each symbol in the figures represents the mean of 2–4 replicates of MDM from one donor using at least three different donors, unless otherwise indicated. Paired $t$-tests were used to compare differences in cell viability. Differences with $p < 0.05$ were considered statistically significant. For differences in gene expression, the WCM Genomics Resource Core Facility calculated the log$_2$-fold change of stimulated cells' transcripts compared to control cells' transcripts using the DESeq2 package (Love et al, 2014), and determined corresponding adjusted $p$-values based on the Benjamini–Hochberg correction for false discovery rate (FDR). Canonical pathway analyses were generated by uploading the log$_2$-fold change of stimulated cells' transcripts compared to control cells' transcripts and the corresponding adjusted $p$-values to the QIAGEN Ingenuity Pathway Analysis (IPA) (Krämer et al, 2013). An "Expression Analysis" using "Expr Log Ratio" as the measurement type was the Core Analysis performed on IPA. The reference gene set was from "Ingenuity Knowledge Base", and the analysis was run using the default settings, except with an adjusted $p$-value cutoff of 0.01.

## Study approval

This study was conducted under a protocol approved by the Institutional Review Board and the Institutional Biosafety Committee of Weill Cornell Medicine (2014-221).

## Data availability

The results of transcriptome sequencing are available under controlled access in the Genotypes and Phenotypes (dbGaP) database with the accession code "phs003607.v1.p1": https://www.ncbi.nlm.nih.gov/projects/gap/cgi-bin/study.cgi?study_id=phs003607.v1.p1.

The source data of this paper are collected in the following database record: biostudies:S-SCDT-10_1038-S44319-024-00171-0.

## Peer review information

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

## Acknowledgements

We thank Dr. Li Zhang for advice; Drs. Li Zhang and Sabine Ehrt for comments on the manuscript; Drs. Kohta Saito, Christopher Brown, and Kathryn Dupnik for drawing blood; Dan Pfau for assistance in isolating monocytes from blood and performing Western blots and an IFN-III reporter assay; Dr. Christopher Brown for clinical strains of Mtb; Dr. Travis Hartman for tdTomato-Mtb; and the WCM Genomics Resources Core Facility for help with RNAseq, especially Adrian Tan for analyzing the RNAseq and Facility Director Dr. Jenny Xiang. This work was supported by NIH grant RO1 AI138940 and by the Abby and Howard P. Milstein Program in Chemical Biology and Translational Medicine. The Department of Microbiology and Immunology is supported by the William Randolph Hearst Trust.

## Author contributions

**Angela M Lee**: Conceptualization; Data curation; Formal analysis; Validation; Investigation; Visualization; Methodology; Writing—original draft. **Carl F Nathan**: Conceptualization; Resources; Supervision; Funding acquisition; Investigation; Project administration; Writing—review and editing.

Source data underlying figure panels in this paper may have individual authorship assigned. Where available, figure panel/source data authorship is listed in the following database record: biostudies:S-SCDT-10_1038-S44319-024-00171-0.

## Disclosure and competing interests statement

The authors declare no competing interests.

# Expanded View Figures

**Figure EV1.   Neutralizing IFN-I Ab mix specifically ablates IFN-I signaling.**

(A) MDM-2 ($n = 1$) were treated with anti-IFNAR1 mAb, anti-IFNAR2 mAb, IgG, or vehicle control (medium) for 2 h, then infected with tdTomato-Mtb. Images were taken at given timepoints post infection at 20X. Scale bars $= 20 \, \mu m$. (B, C) MDM-2 ($n = 1$) were treated with mAbs against IFNAR1, IFNAR2, IgG isotype control, or vehicle control (media) for 2 h, stimulated with IFN-b for 15–20 min, and then lysates were collected for western blot with indicated antibodies. (D) MDM-2 ($n = 1$) were treated with IFN-I neutralizing Ab mixture (see Methods), IgG, or vehicle control (media) for 2 h and stimulated with the indicated concentrations of IFN-b for 15–20 min. Lysates were then collected for western blot with indicated antibodies. (E) MDM-1 ($n = 1$) were treated with IFN-I neutralizing Ab mixture, IgG, or vehicle control (medium) for 2 h and stimulated with IFN-b, IFN-g, or IFN-l for 15–20 min. Lysates were then collected for western blot with indicated antibodies. (F) IFN-III HEKBlue 293 cells ($n = 1$) were treated with IFN-I neutralizing Ab mixture, IgG, or vehicle control (medium) for 2 h and then stimulated with 1000 pg/mL of IFN-l1 (IL-29) overnight. The level of IFN-III signaling was measured via QUANTI-Blue assay. The concentration of IFN-I was estimated by performing a sigmoidal, 4 parameter interpolation from an IL-29 standard curve, using Prism's software. Data Information: Each ● represents the average of 3 technical replicates, while $n$ indicates biological replicates.

▶

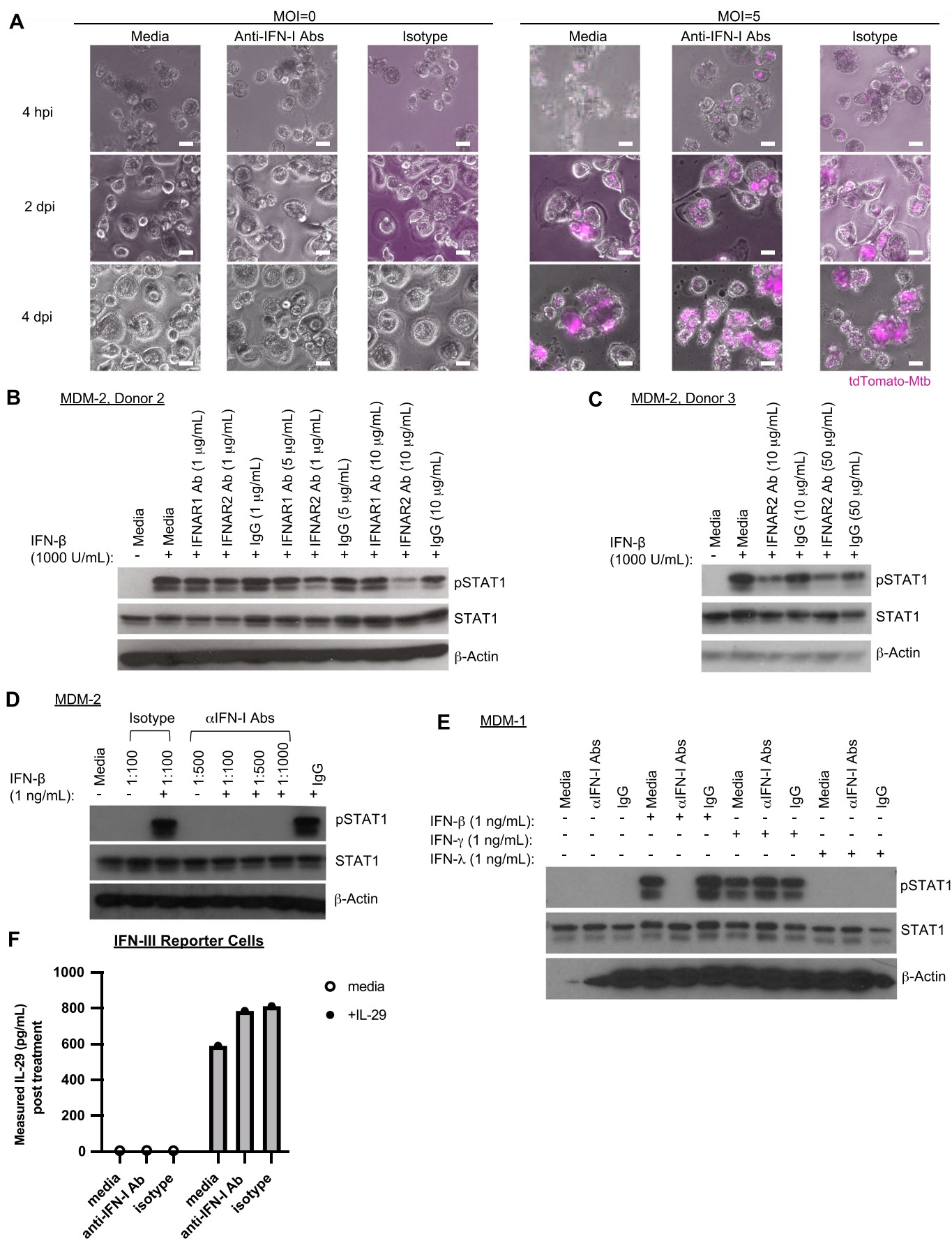

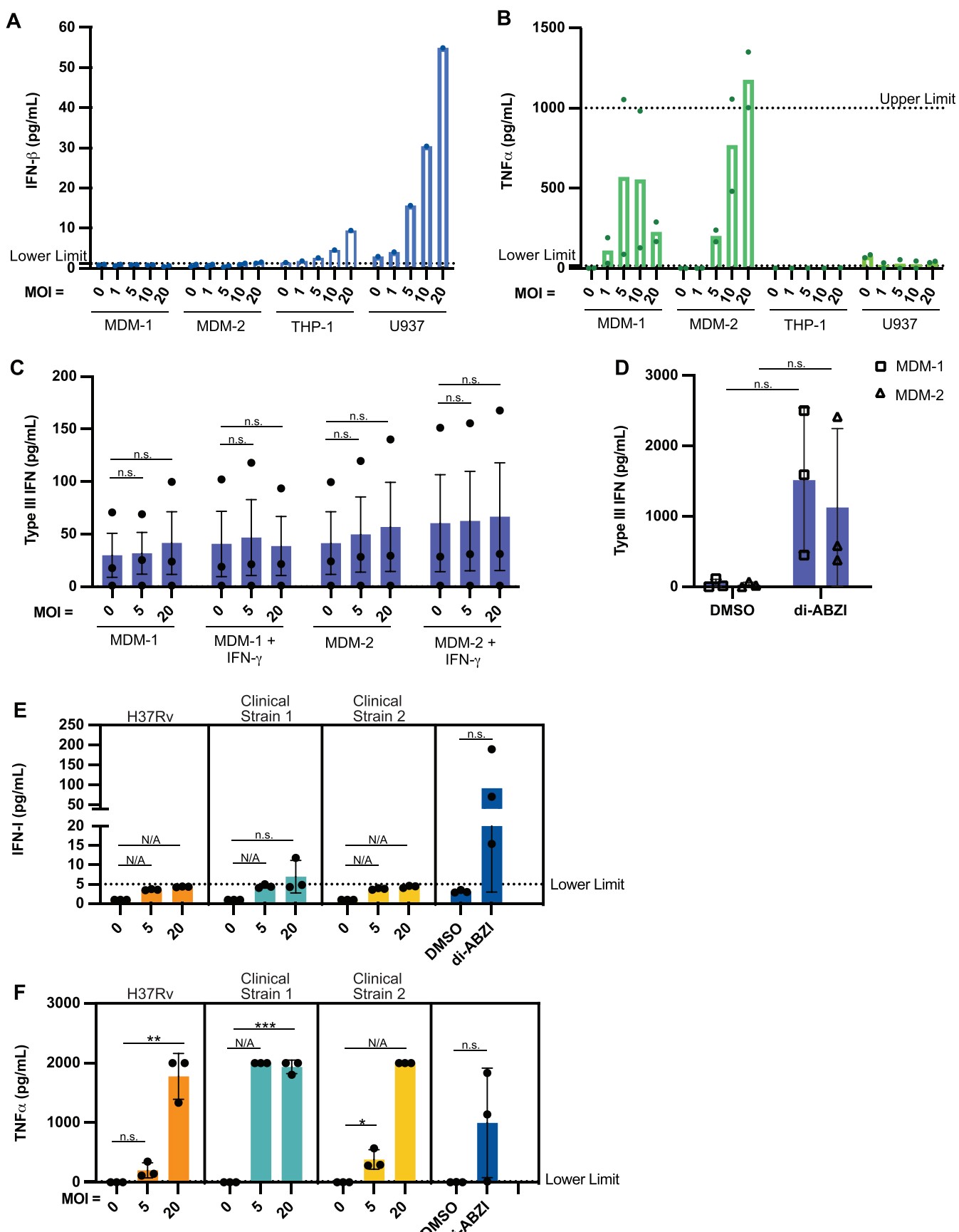

◄ **Figure EV2. Mtb-infected MDMs do not secrete more IFN-III than uninfected MDMs at 1 d post infection nor detectable IFN-I upon infection with clinical isolates of Mtb.**

MDM-1 and MDM-2 ($n = 2$) and 10 ng/mL PMA differentiated THP-1 cells and U937 cells were infected with Mtb H37Rv. 1 d post infection, supernatants were collected. (**A**) IFN-β was measured by a high sensitivity ELISA. (**B**) TNFα was measured by ELISA. (**C, D**) MDM-1 and MDM-2 ($n = 3$) were primed with or without 2.5 ng/mL IFN-γ for 1 day and then (**C**) infected with Mtb strain H37Rv (**C**) or (**D**) treated with 3 μM di-ABZI. Supernatant was collected 1 d post infection. IFN-III expression in the supernatant was measured via the IFN-III HEKBlue reporter assay and quantified by performing a sigmoidal, 4 parameter interpolation from an IFN-λ1 (IL-29) standard curve. (**E, F**) MDM-2 ($n = 3$) were differentiated for only 1 week before infection with Mtb H37Rv, clinical strain 1, clinical Mtb strain 2, or treatment with 3 μM ABZI. (**E**) IFN-I expression in the supernatant was measured via the IFN-I HEKBlue reporter assay, and (**F**) TNFα expression was measured via ELISA. Data Information: (**A–F**) Bar graphs report the mean ± SEM. Each ● represents an individual donor with 2–3 technical replicates per donor. Values too low to be interpolated were assigned a value of 1 pg/mL. Statistical significance was determined using a one-tailed paired, $t$-test (*$p < 0.5$; **$p < 0.01$; ***$p < 0.001$; n.s. indicates no significance; N/A indicates not applicable). N/A was assigned if all values between two groups being tested were below the limit of detection (**E**) or if it was not possible to compute $t$-test because all values per group were the same (**F**). In each panel, $n$ indicates biological replicates.

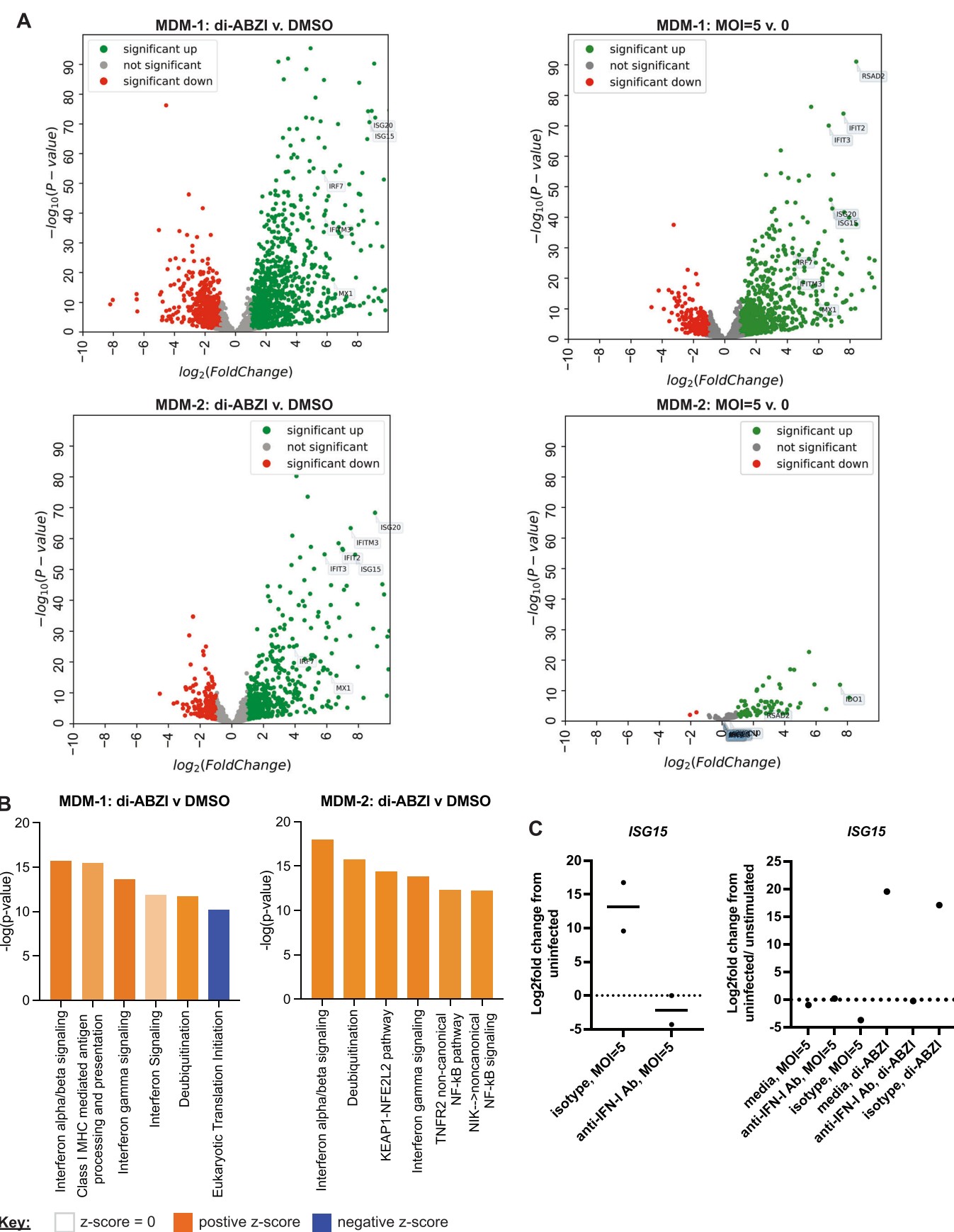

◀ **Figure EV3. Mtb-infected MDM-1 upregulate ISGs in an IFN-I dependent manner.**

(A) MDM-1 and MDM-2 ($n = 3$) were infected with Mtb H37Rv or treated with 3 μM di-ABZI or vehicle controls (equal volume of DMSO as in 3 μM di-ABZI). Lysates were collected 1 day post-treatment or post-infection for RNA sequencing. Volcano plots show $\log_2$fold changes of ISGs determined by Interferome between stimulated/infected MDM-1 compared to unstimulated MDM-1 and the corresponding $-\log_{10}$ adjusted $p$-values. (B) Ingenuity Pathway Analysis (QIAGEN) of the top 6 canonical pathways upregulated in di-ABZI treated MDM-1 (left) and di-ABZI treated MDM-2 (right) determined by –log(adjusted $p$-value) and absolute z-scores >3. (C) Mtb-infected MDM-1 from 2 donors (left) and 1 donor (right) were treated with anti-IFN-I neutralizing Abs or IgG2a isotype Ab for 2 h pre-infection, infected with Mtb H37Rv and again treated with the respective Ab after Mtb was washed out 4 h post infection. Lysates were collected 1 d post infection. *ISG15* expression was measured by RT-qPCR. Data Information: (A) Each ● represents 1 gene. Statistical significance was determined using the DESeq2 package, which uses the Wald test and Benjamini–Hochberg method to correct the FDR (A), and using QIAGEN's IPA analysis, which employs Fisher's exact test and the Benjamini–Hochberg method to control the false discovery rate (Krämer et al, 2013) (B). (C) Each ● represents the average value of 4 technical replicates per donor, while $n$ indicates biological replicates.

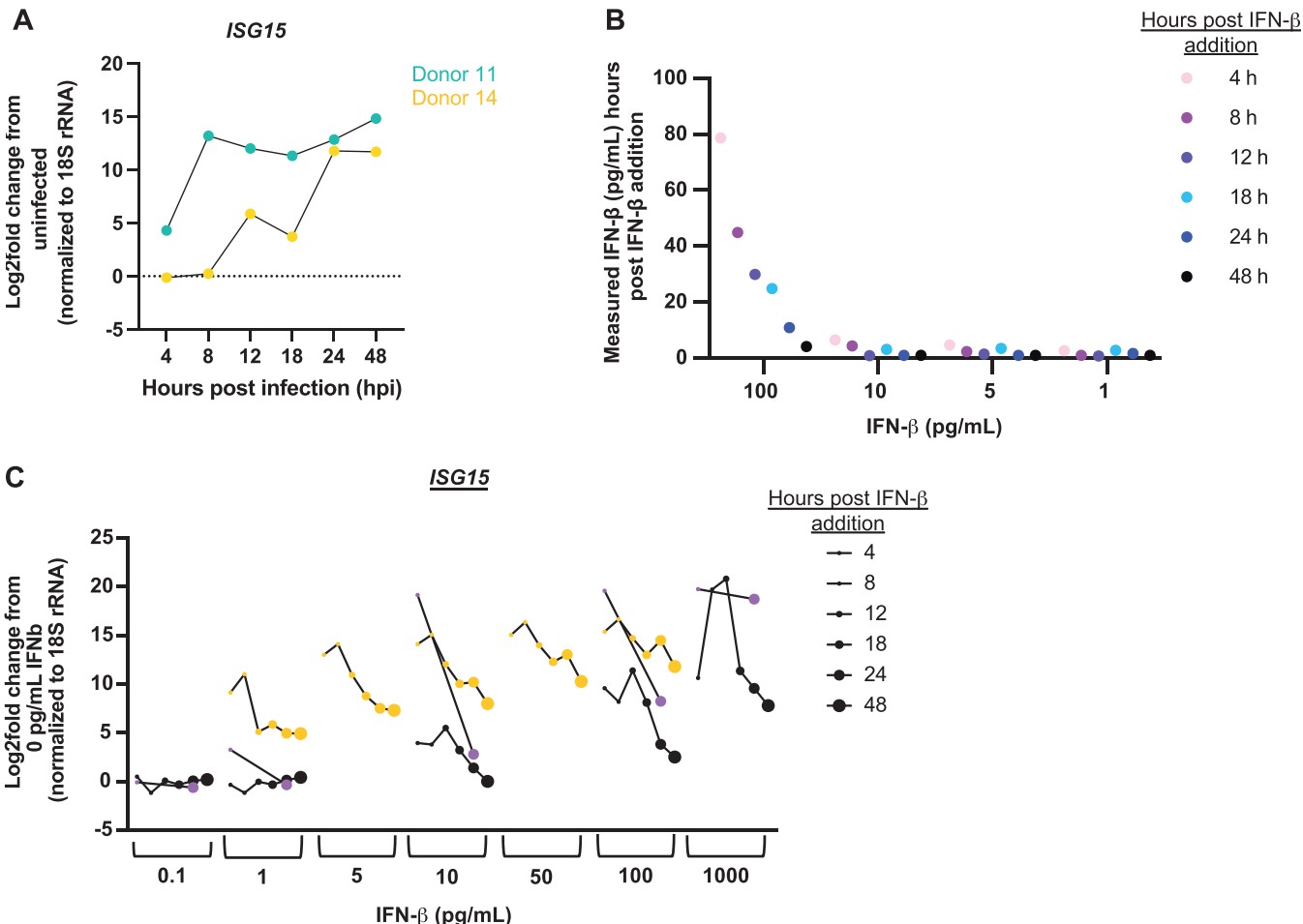

**Figure EV4. MDM-1 consume IFN-β.**

(A) MDM-1 ($n = 2$) were infected with Mtb H37Rv and lysates were collected at indicated timepoints. *ISG15* expression was measured by RT-qPCR. (B) MDM-1 ($n = 1$) were stimulated with 1–100 pg/mL IFN-β, supernatants were collected at indicated timepoints, and IFN-I expression was measured via the IFN-I HEKBlue reporter assay. IFN-I was quantified by performing a sigmoidal, 4 parameter interpolation from an IFN-β standard curve. (C) MDM-1 ($n = 1$–3) were treated with IFN-β and lysates were collected at indicated timepoints. *ISG15* expression was measured by RT-qPCR. Each ● represents the average value of 3–4 technical replicates per donor, while $n$ indicates biological replicates. Data Information: Each ● represents the average value of 3–4 technical replicates per donor (A), the average value of 2 technical replicates per donor (B), and the average value of 3–4 technical replicates per donor (C), $n$ indicates biological replicates.

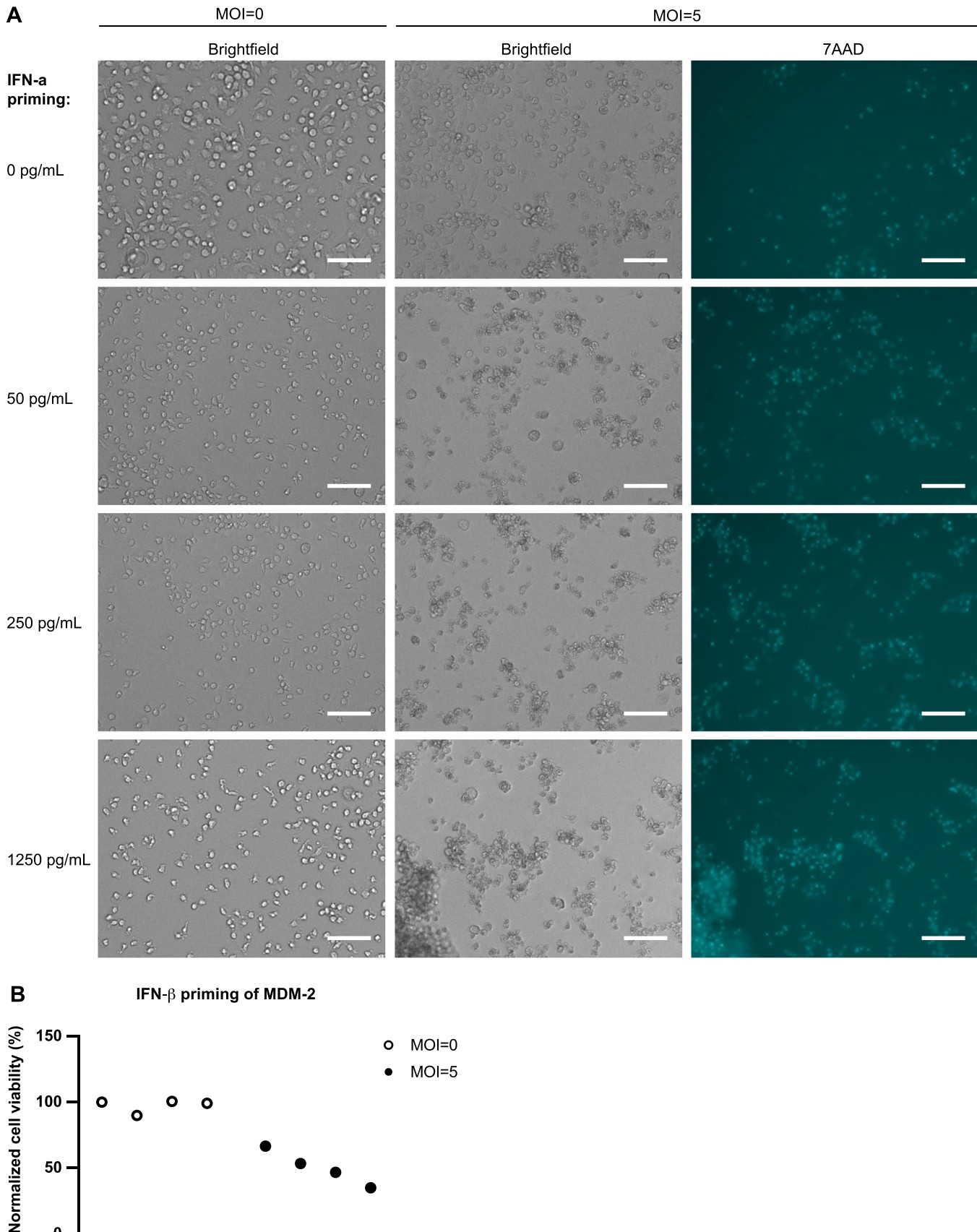

**A**

MOI=0 / MOI=5

IFN-a priming: / Brightfield / Brightfield / 7AAD

0 pg/mL / 50 pg/mL / 250 pg/mL / 1250 pg/mL

**B**

**IFN-β priming of MDM-2**

Normalized cell viability (%)

○ MOI=0
● MOI=5

IFN-β: 0 50 250 1250 0 50 250 1250

◄ **Figure EV5. Exogenous IFN-I exacerbates Mtb-induced MDM death.**

(A) Representative images of MDM-1 ($n = 1$) that were primed with IFN-α 1 day pre-infection, infected with Mtb H37Rv, stained with 7-aminoactinomycin D (7-AAD) and imaged at 10×2 d post infection. Scale bars = 100 μm. (B) MDM-2 ($n = 1$) were primed with IFN-β 1 day pre-infection, infected with Mtb H37Rv, and cell viability was measured via CellTiter Glo assay. Data Information: (B) Dot plot reports mean value of 3 technical replicates per 1 donor, $n$ indicates biological replicates.

