## [Peer Review File · EMBO Reports]

Type I Interferon Exacerbates Mycobacterium tuberculosis Induced Death of Human Macrophages

Angela Lee and Carl Nathan

Corresponding author(s): Carl Nathan (cnathan@med.cornell.edu)

Review Timeline:

Submission Date:	26th Jan 24
Editorial Decision:	1st Feb 24
Revision Received:	21st Feb 24
Editorial Decision:	8th May 24
Revision Received:	21st May 24
Accepted:	29th May 24

Editor: Achim Breiling

Transaction Report: The first review round of this manuscript was performed in another journal.

Dear Dr. Nathan,

Thank you for the submission of your research manuscript to EMBO reports. I now went through your manuscript, the referee reports from a journal outside EMBO press and your revision plan (point-by-point response). The two referees have several comments, concerns, and suggestions to improve the manuscript, indicating that a revision of the manuscript is necessary to allow publication of the study.

Going through your revision plan, it seems that the referee points will be adequately addressed during revision. I thus invite you to revise your manuscript accordingly with the understanding that all concerns must be addressed in the revised manuscript and/or in a final detailed point-by-point response (as indicated in your revision plan). Acceptance of your manuscript will depend on a positive outcome of another round of review using the same referees. I have contacted the other journal, and have obtained referee identities and the original reports. In case the referees do not agree to look into the revised manuscript again, I might involve an expert advisor (arbitrator) to evaluate the revision.

It is EMBO reports policy to allow a single round of major revision only and acceptance of the manuscript will therefore depend on the completeness of your responses included in the next, final version of the manuscript.

- 1) a .docx formatted version of the final manuscript text (including legends for main figures, EV figures and tables), but without the figures included. Figure legends should be compiled at the end of the manuscript text.
- 2) individual production quality figure files as .eps, .tif, .jpg (one file per figure), of main figures (up to 8) and EV figures (up to 5). Please upload these as separate, individual files upon re-submission.

3) a final .docx formatted letter INCLUDING the reviewers' reports and your detailed point-by-point responses to their comments. As part of the EMBO Press transparent editorial process, the point-by-point response is part of the Review Process File (RPF), which will be published alongside your paper.

4) a complete author checklist, which you can download from our author guidelines

(<https://www.embopress.org/page/journal/14693178/authorguide>). Please insert page numbers in the checklist to indicate where the requested information can be found in the manuscript. The completed author checklist will also be part of the RPF.

5) that primary datasets produced in this study (e.g. RNA-seq, CHIP-seq, structural and array data) are deposited in an appropriate public database. If no primary datasets have been deposited, please also state this in a dedicated section (e.g. 'No primary datasets have been generated and deposited'), see below.

The accession numbers and database should be listed in a formal "Data Availability" section (placed after Materials & Methods) that follows the model below. This is now mandatory (like the COI statement). Please note that the Data Availability Section is restricted to new primary data that are part of this study. This section is mandatory. As indicated above, if no primary datasets have been deposited, please state this in this section

Data availability

8) Regarding data quantification and statistics, please make sure that the number "n" for how many independent experiments were performed, their nature (biological versus technical replicates), the bars and error bars (e.g. SEM, SD) and the test used to calculate p-values is indicated in the respective figure legends (also for potential EV figures and all those in the final Appendix). Please also check that all the p-values are explained in the legend, and that these fit to those shown in the figure. Please provide statistical testing where applicable. Please avoid the phrase 'independent experiment', but clearly state if these were biological or technical replicates. Please also indicate (e.g. with n.s.) if testing was performed, but the differences are not significant. In case n=2, please show the data as separate datapoints without error bars and statistics. See also: <http://www.embopress.org/page/journal/14693178/authorguide#statisticalanalysis>

Please format the figure legends for main, EV and Appendix figures) according to our journal style. See the respective section in our guide to authors (please find the link below). Please separate each panel description by a line brake and make sure that the panels are listed in alphabetic order. Moreover, please add to each legend a 'Data Information' section explaining the statistics used or providing information regarding replicates and scales.

9) Please also note our reference format:

10) We updated our journal's competing interests policy in January 2022 and request authors to consider both actual and perceived competing interests. Please review the policy <https://www.embopress.org/competing-interests> and update your competing interests if necessary. Please name this section 'Disclosure and Competing Interests Statement' and put it after the Acknowledgements section.

11) We now use CRediT to specify the contributions of each author in the journal submission system. CRediT replaces the

author contribution section. Please use the free text box to provide more detailed descriptions and do not provide an author contributions section in the revised manuscript text file. See also guide to authors:

<https://www.embopress.org/page/journal/14693178/authorguide#authorshipguidelines>

12) Please add scale bars of similar style and thickness to all the microscopic images, using clearly visible black or white bars (depending on the background). Please place these in the lower right corner of the images themselves. Please do not write on or near the bars in the image but define the size in the respective figure legend.

13) Please provide a final title with not more than 100 characters (including spaces).

14) Please provide the abstract written in present tense throughout and with not more than 175 words.

15) Please make sure that all the funding information is also entered into the online submission system and that it is complete and similar to the one in the acknowledgement section of the manuscript text file.

16) We would encourage you to use 'Structured Methods', our new Materials and Methods format. According to this format, the Materials and Methods section should include a Reagents and Tools Table (listing key reagents, experimental models, software and relevant equipment and including their sources and relevant identifiers), uploaded as separate file, followed by a Methods and Protocols section in which we encourage the authors to describe their methods using a step-by-step protocol format with bullet points, to facilitate the adoption of the methodologies across labs. More information on how to adhere to this format as well as downloadable templates (.doc or .xls) for the Reagents and Tools Table can be found in our author guidelines (section 'Structured Methods'):

17) Please order the manuscript sections like this, using these names:

Title page - Abstract - Keywords - Introduction - Results - Discussion - Materials and Methods - Data availability section - Acknowledgements - Disclosure and Competing Interests Statement - References - Figure legends - Expanded View Figure legends

Please note that all corresponding authors are required to supply an ORCID ID for their name upon submission of a revised manuscript. Please find instructions on how to link the ORCID ID to the account in our manuscript tracking system in our Author guidelines: <http://www.embopress.org/page/journal/14693178/authorguide#authorshipguidelines>

I look forward to seeing a revised version of your manuscript when it is ready. Please let me know if you have questions or comments regarding the revision.

Best,

Achim Breiling
Senior editor
EMBO reports

Dear Dr. Achim Breiling,

Thank you for giving us the opportunity to submit a revised version of the manuscript, "Type I Interferon Exacerbates *Mycobacterium tuberculosis* induced human macrophage death," for publication at EMBO Reports. We appreciate the time and effort that you and the reviewers have given to provide thoughtful suggestions to improve our paper. We have responded to all of the suggestions that the reviewers have given, with changes underlined in the manuscript. Please see below of the point-by-point response, in blue, to the reviewer's comments.

Reviewer #1:

Lee and Nathan present a compelling manuscript on establishing the role of type I interferon in primary human monocyte-derived macrophages as models for *Mycobacterium tuberculosis* infection. Type I interferon signatures are a hallmark of TB disease, but their roles and impact on Mtb-infected macrophages remains poorly understood. The manuscript tackles two important questions in the field: the first is the secretion, signaling dynamics and functional consequences of type I interferons during Mtb infection in commonly used in vitro monocyte-derived macrophage (MDM) culture systems. The second is comparing two of these widely used methods to generate MDMs, since the heterogeneity of established methods to generate these MDMs have added a lot of complexity in interpreting the data in the TB field. The dogma is that these downstream effects, which include cell death (and possible subsequent promotion of Mtb dissemination) and the induction of interferon-stimulated genes (ISG), are mediated through phospho-STAT1 signaling. It remains unclear if infected macrophages themselves secrete IFN, and the kinetics of IFNAR activation to mediate the downstream function of IFN. The authors dissect these steps in detailed experimental steps using two MDM models and an impressive suite of tools to test the function and signaling kinetics of type I IFN. I invite the authors to address the following points to more definitively clarify the points made in the manuscript:

1. In all experiments, the macrophages are treated as if all the cells in the well are infected. We cannot definitively distinguish induction of IFN α or beta itself, or ISGs from Mtb-infected cells compared to bystander cells in the well. Even at high MOIs you expect many cells to remain uninfected, which may be the source of IFN or IFN-responsive cells. This distinction can only be made using a reporter Mtb strain, are separating Mtb-infected cells from those which are uninfected. Can you perform an experiment where you sort infected and uninfected MDMs, to test induction of ISGs by qPCR. At minimum, soften the language in the discussion to include the possibility that the secretion of IFN and induction of ISGs may be mediated by either the infected or bystander cells, which is relevant to the situation in vivo, as demonstrated in Kotov et al, 2023.

We thank Reviewer 1 for the suggestion to clarify if IFN-I secretion and/or induction of ISGs are taking place in Mtb-infected or bystander cells. Unfortunately, we do not have a flow cytometer

within the bsl3 facility. While it might be possible to fix samples, bring them out of the bsl3, flow sort for a fluorescent marker expressed by Mtb, extract RNA, and then do qPCR, this would be a new technique in the lab that would be time consuming and costly to troubleshoot and perform. We have taken Reviewer 1's suggestion to note in the Discussion (page 12-13, lines 259-268) the possibility that the secretion of IFN-I and induction and ISG may be occurring in either infected or bystander cells.

2. In Figure 1A and B, you state that you're showing ATP levels as a measure of viability, but that's not what's shown on the y-axis. Report the ATP levels as indicated in the text (first paragraph of results).

We thank Reviewer 1 for pointing this out. We report the normalized cell viability in order to adjust for the variabilities among donors. We have added text in the Figure 1 legend and in 1st paragraph of the Results (page 6, line 114-115) to indicate that the y-axis is displaying normalized cell viability.

3. Experiment in figure 2A: are these independent donors or technical replicates? Please state in text- and if they are independent donors, you can use a different symbol shape to track whether different donors have consistently higher or lower cytokine secretion levels.

Each dot represents results for an individual donor averaged from 3 technical replicates. In Figure 2A, we have now colored each symbol using one color per donor. The figure legend of Figure 2A has now been corrected to state that each symbol represents 1 donor, with 3 donors in Figure 2A (page 28, line 659-661).

4. Figure S2E, the use of different clinical strains is impactful. However, the classification of low and high transmissibility based on the original paper is debatable. Soften the language to say these are two different clinical isolates described previously.

In keeping with EMBO Reports format, Figure S2E is now Figure EV2E. We have removed the classification of low and high transmissibility and renamed the strains as "clinical strain 1" and "clinical strain 2" as Reviewer 1 has suggested (page 7-8, lines 146-150).

5. In figure 3B, add pathway enrichment scores for di-ABZI to the graph. This will resolve IFN-signaled STING-mediated pathways compared to those induced by Mtb alone.

Thank you for this advice. For space constraints reasons in Figure 3B, we have added the pathway enrichment score for di-ABZI of both MDM types in Fig. EV3B and added a statement that IFN-I signaling was the top pathway upregulated in the di-ABZI treated MDM-1 and MDM-2 (page 8, line 166-167).

6. What's the rationale for quantifying [RNA] in figure 4B?

To extend our comparison of the two methods for differentiating MDM, we illustrated their

striking morphologic differences. In the course of measuring RNA to synthesize similar quantities of cDNA for qPCR across samples, we were struck by their differences in RNA content as well and we think this is worth reporting.

7. For experiments in figure 5, it's possible as the authors indicated that IFN-I was rapidly bound to IFNAR and thus not detectable in the media. The prediction would be that IFNAR^{-/-} MDMs would retain higher IFN-I in the media. While I acknowledge the complexity of genetically editing these primary human MDMs, is there precedence for longer IFN half-life in IFNAR^{-/-} mice or cell lines, or in people with loss of function mutations in IFNAR? This will rule out whether the issue is rapid internalization and signaling, or degradation due to a short IFN half-life. Also the authors cite the possibility of autocrine signaling, has that been shown in prior studies?

We have not found literature bearing on the half-life of IFN-I in IFNAR^{-/-} mice or cell lines or in people with loss of function mutations in IFNAR. It would be an interesting experiment, but beyond our means, to generate IFNAR^{-/-} primary human MDM at high efficiency and to document the per-cell success of the knockouts so as to be able to interpret the results with confidence. Nor have we found reports of autocrine signaling by IFN-I.

Reviewer #2:

This study serves as a continuation of a prior report by the same group (Zhang et al. 2022 JEM), where the authors demonstrated that type I IFN enhances Mtb-induced cell death in murine macrophages.

In their investigation, the authors examine whether M. tuberculosis triggers the secretion of type I IFN in human macrophages and if this secretion results in macrophage death, as previously reported in mouse macrophages by the authors themselves in a prior study. Two macrophage models are employed: "MDM1", following a widely used protocol (Novikov et al. 2011 J Immunol) or variations thereof, and "MDM2", following an original protocol (Vogt and Nathan 2011 J Clin Invest) leading to macrophages with enhanced mycobactericidal activity.

According to their findings:

- The death of human macrophages (MDM1 or MDM2) infected with Mtb appears to be independent of type I IFN. Blocking both type I IFN and the type I IFN receptor IFNAR with a cocktail of antibodies before and during infection shows no effect on cell death at 2 or 4 days post-infection, irrespective of the presence or absence of gamma interferon.
- MDM1 and MDM2 macrophages do not secrete type I IFN on the first day after infection, as demonstrated by two independent methods: a reporter cell line and an ultrasensitive ELISA.
- MDM1 macrophages, but not MDM2, exhibit an increase in ISG during Mtb infection in a type I IFN-dependent manner.
- The absence of ISG induction in MDM2 appears linked to specific aspects of the cell differentiation protocol, specifically the use of human plasma instead of FBS and the duration of differentiation.

- MDM1 infected with Mtb indeed produce type I IFN, but not until 48 hours after infection. However, ISG induction is noticeable earlier, at 8-12 hours after infection. This suggests that infected macrophages may produce a low level of IFN during the 0-24 h time window, with the cytokine potentially being cleared from the medium at a similar rate via endocytosis through IFNAR.
- Finally, exogenous type I IFN added to Mtb-infected macrophages induces cell death, and this effect is observed only when the cells are infected.

The addressed question in this study holds significant relevance as type I IFN plays a crucial role in the pathogenesis of human TB. The manuscript is well-crafted, and the experiments have been meticulously conducted, with conclusions grounded in robust data that are not overly interpreted.

Nevertheless, in its current state, the study appears more as a compilation of intriguing observations that lack clear interconnection. This leaves the reader without a comprehensive understanding of whether Mtb-infected macrophages produce type I IFN *in vivo* during human TB, and if the production of type I IFN, whether by infected macrophages or other cell types, induces the death of infected macrophages in patients.

Considering the complexity of type I IFN signaling, which involves dynamic aspects in IFN production and reuptake, as well as in ISG induction, and encompasses a blend of exogenous and endogenous signaling via membrane and endosomal IFNAR, a more cohesive exploration may be necessary to establish a clearer picture. The investigation seems to be in its early stages when delving into this intricate process. Numerous intriguing observations have been made, some of which even appear paradoxical-such as the generation of ISG in the absence of detectable IFN-yet there is a lack of mechanistic explanation for these phenomena. Moreover, it is improbable that the two macrophage models employed in this study entirely replicate human lung macrophages. The mechanisms connecting Mtb infection to IFN production, ISG induction, and cell death in TB patients may be distinctly different, and this aspect remains unexplored.

To arrive at a more informative conclusion, it may be necessary to conduct a more comprehensive and mechanistic investigation utilizing primary human alveolar macrophages.

We appreciate Reviewer 2's constructive critique. Indeed, determining the mechanism for the generation of ISG in the absence of detectable IFN-I requires further investigation, despite our best efforts to answer these questions over the past 5 years. While alveolar macrophages are the primary niche of Mtb in the first 2 weeks of infection, recruited MDM are the primary niche of Mtb during chronic infection and thus serve as a better model of determining the role of IFN-I in macrophages during Mtb activation/reactivation from latency. Furthermore, obtaining primary human alveolar macrophages is an invasive procedure that would require an IRB protocol and clinical collaborators. With more time and resources, it would be interesting to understand the role of IFN-I in the death of Mtb-infected primary human macrophages to model early Mtb infection of humans, as Reviewer 2 suggests. However, the grant supporting this work ended six months ago. The first author has completed her PhD studies and is moving on. We hope it will be helpful to the research community to report the present results, which may encourage others to continue investigating these questions.

Dear Prof. Nathan,

Thank you for the submission of your revised manuscript to our editorial offices. I have already forwarded to you the report I have received from the referee that I asked to re-evaluate the study, you will find again below. As you know, the referee indicated that s/he is not yet satisfied with the revision and has remaining concerns. I went through your rebuttal letter and asked the referee for comments. The referee feels that your response addresses his/her points, except for the first point. The referee states:

"I still think that raw viability should be displayed. The data shown [in the rebuttal] exactly demonstrates my concern that the normalized data eclipses the heterogeneity of basal ATP levels per donor sample. I think the normalization is misleading."

I thus ask you to address this point (as indicated by the referee) and the remaining points (as indicated in your rebuttal) in a final revised manuscript. Please also provide a final p-b-p-response with your resubmission, addressing the remaining the referee points.

- Please make sure that the number "n" for how many independent experiments were performed, their nature (biological versus technical replicates), the bars and error bars (e.g. SEM, SD) and the test used to calculate p-values is indicated in the respective figure legends (also for potential EV figures and all those in the final Appendix). Please also check that all the p-values are explained in the legend, and that these fit to those shown in the figure. Please provide statistical testing where applicable. Please avoid the phrase 'independent experiment', but clearly state if these were biological or technical replicates. Please also indicate (e.g. with n.s.) if testing was performed, but the differences are not significant. In case n=2, please show the data as separate datapoints without error bars and statistics. See also:
<http://www.embopress.org/page/journal/14693178/authorguide#statisticalanalysis>

If n<5, please show single datapoints for diagrams. Presently, some diagrams seem to show only partial or no statistics, or the n.s. is missing. Please check. Moreover:

- Please note that the legends for figures 5b-c is incorrectly labelled as 5c-d. This needs to be rectified.
- Please note that figure EV 1f seem not contain any statistical parameters, kindly rectify the statistics test related information in the figure legend appropriately.
- Please indicate the statistical test used for data analysis in the legends of figure 3b; EV 3a-b.
- Please note that in figures 4b-c; EV 2f; there is a mismatch between the annotated p values in the figure legend and the annotated p values in the figure file that should be corrected.

- Please add to each legend (main and EV figures, where applicable) a 'Data Information' section explaining the statistics used or providing information regarding replicates and scales. See:

- Please name the materials and methods section just 'Methods'.
- Please name "Table I" "Table 1" and place it between the main and EV figure legends.
- Please update the "Data Availability" section, adding the accession IDs for the Genotypes and Phenotypes (dbGaP) database and direct links to the deposited data. Please make sure that the data are public latest upon online publication of the study.
- Please complete the author checklist adding the information regarding the deposited datasets (cells D112 and D113).
- Thanks for providing the requested source data (SD). Please make sure the final SD is uploaded as one folder per figure (ZIPed up together) and the SD for EV figures and Appendix data uploaded ZIPed together in a separate folder. Moreover, while analysing the SD, we noted some source data files contain duplicated values that seem to come from different measurements/conditions/donors/replicates (please see the files in the attached ZIP folder). Please look through these and check if the numbers are correct or please provide explanations in the final point-by-point response.

In addition, I would need from you:

Best,

Referee #1:

Although the authors were somewhat responsive to the previous round of critiques, there are still critical areas in need of revision to ensure reliability of conclusions and make the display of the data more transparent.

- Please refrain from using "normalized viability". I understand the logic of normalizing to the uninfected background, but showing the raw viability in both conditions is the most transparent way to show the data. Right now the data display eclipses background levels (e.g. if you're starting with cells with poor quality, are they more sensitive to IFN manipulation)?
- The use of technical replicates of the same donors does not suffice to establish generalizability. The conclusions drawn from figure 2 about IFN secretion are too important to draw from a single donor. The reality with human samples is that there are very heterogeneous and biological replicates are indispensable to draw such important conclusions about source of IFN. A single biological replicate is simply not enough.
- The response to the logic of showing RNA concentrations is still weak. Is there a biological significance to differences in RNA content?

Dear Dr. Breiling,

Thank you for giving us the opportunity to submit a second revised version of the manuscript, "Type I Interferon Exacerbates *Mycobacterium tuberculosis* induced human macrophage death," for publication at EMBO Reports. We appreciate the time and effort that you and the Referee have given to provide further suggestions and corrections to improve our paper. We have addressed all of the suggestions that you and the Referee have given, with changes underlined in the manuscript. Please see below of the point-by-point response, in blue, to the Referee's and your comments.

Referee #1:

Although the authors were somewhat responsive to the previous round of critiques, there are still critical areas in need of revision to ensure reliability of conclusions and make the display of the data more transparent.

- Please refrain from using "normalized viability". I understand the logic of normalizing to the uninfected background, but showing the raw viability in both conditions is the most transparent way to show the data. Right now the data display eclipses background levels (e.g. if you're starting with cells with poor quality, are they more sensitive to IFN manipulation)?

While we agree with Referee #1 that the raw data is transparent and should be made available, the reason we normalized the viability to the uninfected controls is because donors can show large variability (as Referee #1 noted in the next point). For example, below we show the raw data, which is the relative light units of luminescence based on ATP levels (left figure). One donor's MDM depicted in yellow had much less ATP at baseline (Media, MOI=0) than the other donors. When the raw data is shown graphically, differences in ATP levels between different conditions of the yellow donor's MDM cannot be seen (left figure). When cell viability is normalized to baseline (right figure), differences in cell viability between different conditions of the yellow donor's MDM can be seen. Thus, normalized viability better shows differences based on how the MDM were treated rather than on donor differences pre-existing before infection with Mtb. We have added an explanation in the Methods to indicate why the normalized viability was used.

It is not always the case that donors display as large differences in cell viability. For example, below we show that the raw data (left figure) closely resembles the normalized results (middle figure). The major difference between these two figures (below left and middle figures) is the scale of the y-axis values. In this case, all of donors have similar baselines (MOI=0, 0 pg/mL of IFN-a), and thus there are no major differences seen between the raw and normalized data.

IFN-a priming of uninfected MDM did not have an effect on MDM viability (right and middle figures). In addition, IFN-b priming or IFN-a/b addition did not affect uninfected MDM viability (Figure 6A-D), indicating that cells from different donors were not more sensitive to IFN-I treatment when not infected with Mtb.

The reason why the cell viability fold change is shown in Figure 6 is because Mtb infection kills MDM even in the absence of added IFN-I (left and middle figure) and the effects of IFN-I on Mtb infected MDM are partially masked. In order to better see whether IFN-I treatment leads to greater MDM death during Mtb infection, we showed the fold change of the cell viability between IFN-I treated and non-IFN-I treated in MOI=0 group and then the fold change of the cell viability between IFN-I treated and non-IFN-I treated in MOI=5 Mtb infected group, respectively (right figure).

"I still think that raw viability should be displayed. The data shown [in the rebuttal] exactly demonstrates my concern that the normalized data eclipses the heterogeneity of basal ATP levels per donor sample. I think the normalization is misleading."

We displayed the raw values for viability in the Appendix for full inspection of the data. We believe it is important to keep the normalized data in the main figures, because the results that we are demonstrating are the effects that IFN-I have on the cell viability of Mtb infected MDM, which are independent of the heterogeneity of basal ATP levels per donor before addition of IFN-I. It is well known that cells from human donors display heterogeneity at basal levels; it is not a new point that deserves to be the central conclusion of this paper. We are focused on displaying the consistent pattern of responses of cells from different donors, not the basal heterogeneity of different donors' cells before application of the stimulus to which they respond.

In the right figure below displaying the raw data, the yellow donor's MDMs cell viability in response to anti-IFN-I Ab or isotype Ab cannot be appreciated from this display. Furthermore, for the yellow donor, it looks like there is no difference in cell viability between no infection and Mtb infection (MOI=0 vs. MOI=20) (right figure). When

the data is normalized, the cell viability of the yellow donor's MDM can be seen to decrease with Mtb infection to a similar extent as the other 2 donors (left figure). Thus, the raw data (right figure) makes it seem like the yellow donor's MDM are already dead to begin with and that neither Mtb infection nor inhibition of IFN-I makes a difference in cell viability. The normalized data (left figure) shows that the yellow donor's MDM are not all dead and that for all 3 donors Mtb infection kills the MDMs at a similar rate but that inhibition of IFN-I does not affect cell viability. Therefore, without ability to show normalized data, the central observation is difficult to convey to the reader.

Moreover, we respectfully point out that displaying normalized cell viability data in reference to a negative control upon drug treatment is widely used in pharmacology (<https://www.altex.org/index.php/altex/article/download/825/838/2132#:~:text=To%20make%20such%20data%20easily,the%20system%20behavior%20are%20determined>), and normalized cell viability data upon IFN-I treatment/inhibition has been already been published for Mtb infected mouse macrophages in the Journal of Experimental Medicine (<https://rupress.org/jem/article/218/2/e20200887/211515/Type-I-interferon-signaling-mediates-Mycobacterium>).

- The use of technical replicates of the same donors does not suffice to establish generalizability. The conclusions drawn from figure 2 about IFN secretion are too important to draw from a single donor. The reality with human samples is that there are very heterogeneous and biological replicates are indispensable to draw such important conclusions about source of IFN. A single biological replicate is simply not enough.

We agree that humans are very heterogeneous and that biological replicates are important. Figure 2A shows 3 donors (2-3 technical replicates were done per donor). We have added 'n=3' to the figure legend to clarify the number of donors in Figure 2A. Additionally, in Figure EV2E, we also show that MDM from 3 donors do not secrete increased IFN-I when infected with the laboratory Mtb strain, H37Rv. 2 of these 3 donors are different from the donors in Figure 2A. Thus, in the manuscript there are a total of 5 donors under the same conditions whose cells do not secrete increased IFN-I upon H37Rv infection. Furthermore, Figure 2E shows that MDM from 3 more donors do not secrete IFN-I upon H37Rv and DMSO treatment above the limit of detection, indicating that a total of 8 donors do not secrete increased IFN-I upon H37Rv infection under slightly varying conditions.

- The response to the logic of showing RNA concentrations is still weak. Is there a biological significance to differences in RNA content?

Presumably, Referee #1 is referring to Fig. 4C? There is no biological significance other than the possibility that either there is more transcription or more rRNA present in the MDM differentiated in human plasma. More experiments would be required to determine the biological significance of the differences in RNA content. We moved this figure to Appendix Fig. S2B.

- Please make sure that the number "n" for how many independent experiments were performed, their nature (biological versus technical replicates), the bars and error bars (e.g. SEM, SD) and the test used to calculate p-values is indicated in the respective figure legends (also for potential EV figures and all those in the final Appendix). Please also check that all the p-values are explained in the legend, and that these fit to those shown in the figure. Please provide statistical testing where applicable. Please avoid the phrase 'independent experiment', but clearly state if these were biological or technical replicates. Please also indicate (e.g. with n.s.) if testing was performed, but the differences are not significant. In case n=2, please show the data as separate datapoints without error bars and statistics. See also: <http://www.embopress.org/page/journal/14693178/authorguide#statisticalanalysis>

- We have stated that n indicates biological replicates in the figure legends, and we have stated the relevant technical and biological replicates

If n<5, please show single datapoints for diagrams. Presently, some diagrams seem to show only partial or no statistics, or the n.s. is missing. Please check. Moreover:

- Please note that the legends for figures 5b-c is incorrectly labelled as 5c-d. This needs to be rectified.

- We have rectified this mistake.

- Please note that figure EV 1f seem not contain any statistical parameters, kindly rectify the statistics test related information in the figure legend appropriately.

- We deleted the statistical test related information from the figure legend of EV1f.

- Please indicate the statistical test used for data analysis in the legends of figure 3b; EV 3a-b.

- We have indicated the statistical tests used for data analysis in the legends of figure 3b; EV 3a-b.

- Please note that in figures 4b-c; EV 2f; there is a mismatch between the annotated p values in the figure legend and the annotated p values in the figure file that should be corrected.

- We have corrected the mismatched annotation of p-values for figures 4b-c; EV 2f.

- Please add to each legend (main and EV figures, where applicable) a 'Data Information' section explaining the statistics used or providing information regarding replicates and scales. See:

- We have added a Data Information section for each figure legend.

- Please name the materials and methods section just 'Methods'.
- 'Methods' was already labelled 'Methods'

- Please name "Table I" "Table 1" and place it between the main and EV figure legends.
- We have renamed 'Table 1' to 'Table I,' and placed it between the main and EV figure legends.

- Please update the "Data Availability" section, adding the accession IDs for the Genotypes and Phenotypes (dbGaP) database and direct links to the deposited data. Please make sure that the data are public latest upon online publication of the study.
- We have added the accession ID and direct link to the dbGaP database. The data is now online under controlled access, which is required for human data.

- Please complete the author checklist adding the information regarding the deposited datasets (cells D112 and D113).
- The information regarding cells D112 and D113 in the author checklist has been updated.

- Thanks for providing the requested source data (SD). Please make sure the final SD is uploaded as one folder per figure (ZIPed up together) and the SD for EV figures and Appendix data uploaded ZIPed together in a separate folder. Moreover, while analysing the SD, we noted some source data files contain duplicated values that seem to come from different measurements/conditions/donors/replicates (please see the files in the attached ZIP folder). Please look through these and check if the numbers are correct or please provide explanations in the final point-by-point response.
- The source data and the source data checklist has been updated. We were under the impression that the source data can only be linked to the main figures with the source data checklist highlighting the 6 main figures.

In addition, I would need from you:

- a short, two-sentence summary of the manuscript (not more than 35 words).
- two to four short (!) bullet points highlighting the key findings of your study (two lines each).
 - Please find a short two-sentence summary and the 4 bullet points attached.

- a schematic summary figure as separate file that provides a sketch of the major findings (not a data image) in jpeg or tiff format (with the exact width of 550 pixels and a height of not more than 400 pixels) that can be used as a visual synopsis on our website.
 - Please find a schematic attached.

Prof. Carl Nathan
Weill Cornell Medical College
Microbiology & Immunology
1300 York Avenue
New York, New York 10065
United States

Dear Prof. Nathan,

I am very pleased to accept your manuscript for publication in the next available issue of EMBO reports. Thank you for your contribution to our journal.

Yours sincerely,
